# Development of Chitosan/Silver Sulfadiazine/Zeolite Composite Films for Wound Dressing

**DOI:** 10.3390/pharmaceutics11100535

**Published:** 2019-10-14

**Authors:** Patricia Hissae Yassue-Cordeiro, Cássio Henrique Zandonai, Bianca Pereira Genesi, Patrícia Santos Lopes, Elena Sanchez-Lopez, Maria Luisa Garcia, Nádia Regina Camargo Fernandes-Machado, Patrícia Severino, Eliana B. Souto, Classius Ferreira da Silva

**Affiliations:** 1Department of Chemical Engineering, Universidade Tecnológica Federal do Paraná, Av. dos Pioneiros, 3131, Jardim Morumbi, Londrina-PR 86036-370, Brazil; pcordeiro@utfpr.edu.br (P.H.Y.-C.); nrcfmachado@uem.br (N.R.C.F.-M.); 2Department of Chemical Engineering, Universidade Estadual de Maringá, Av. Colombo 5790 Bloco D-90, Maringá 87020-900, Brazil; chzandonai@gmail.com; 3Instituto de Ciências Ambientais, Químicas e Farmacêuticas, Universidade Federal de São Paulo, Rua São Nicolau 210, Diadema 09913-030, Brazil; bianca.genesi@hotmail.com (B.P.G.); patslopes@hotmail.com (P.S.L.); 4Department of Pharmacy, Pharmaceutical Technology and Physical Chemistry, Faculty of Pharmacy and Food Sciences, University of Barcelona, Av. Joan XXIII 27–31, 08028 Barcelona, Spainmarisagarcia@ub.edu (M.L.G.); 5Institute of Nanoscience and nanotechnology (IN2UB). Faculty of Pharmacy, University of Barcelona, 08028 Barcelona, Spain; 6CIBERNED Centros de Biomedicina en Red de Enfermedades Neurodegenerativas, Facultat de Farmàcia, Universitat de Barcelona, 08028 Barcelona, Spain; 7Instituto de Tecnologia e Pesquisa, Universidade Tiradentes, Aracaju 49010-390, Brazil; pattypharma@gmail.com; 8Tiradentes Institute, 150 Mt Vernon St, Dorchester, MA 02125, USA; 9Faculdade de Farmácia, Universidade de Coimbra, Pólo das Ciências da Saúde, Azinhaga de Santa Comba, 3000-548 Coimbra, Portugal; 10CEB—Centro de Engenharia Biológica, Universidade do Minho, Campus de Gualtar, 4710-057 Braga, Portugal

**Keywords:** wound dressing, silver sulfadiazine, zeolites, chitosan, composite films

## Abstract

Biopolymeric films with silver sulfadiazine (AgSD) are proposed as an alternative to the occlusive AgSD-containing creams and gauzes, which are commonly used in the treatment of conventional burns. While the recognized cytotoxicity of AgSD has been reported to compromise its use as an antimicrobial drug in pharmaceuticals, this limitation can be overcome by developing sustained-release formulations. Microporous materials as zeolites can be used as drug delivery systems for sustained release of AgSD. The purpose of this work was the development and characterization of chitosan/zeolite composite films to be used as wound dressings. Zeolite was impregnated with AgSD before the production of the composite films. The physicochemical properties of zeolites and the films were evaluated, as well as the antimicrobial activity of the polymeric films and the cytotoxicity of the films in fibroblasts Balb 3T3/c. Impregnated zeolite exhibited changes in FTIR spectra and XRD diffraction patterns, in comparison to non-impregnated composites, which corroborate the results obtained with EDX-SEM. The pure chitosan film was compact and without noticeable defects and macropores, while the film with zeolite was opaquer, more rigid, and efficient against *Candida albicans* and some gram-negative bacteria. The safety evaluation showed that although the AgSD films present cytotoxicity, they could be used in a concentration-dependent fashion.

## 1. Introduction

A burn is a wound of any traumatic type that compromises the function of the epithelial tissue. It is considered a significant problem, not only for the gravity when acute, but also concerning its significant sequelae that may forever mark burned patients [1]. This type of injury is distinguished from others due to the high risk of colonization by pathogenic bacteria, the presence of large amounts of non-viable tissue, the loss of a large quantity of water and blood, risk to remain open for extended periods of time until its complete healing, and frequently need to mobilize tissue for wound closure [2,3].

Infection is one of the most frequent and severe complications of burn patients, being responsible for 75–80% of deaths worldwide. A local indication of infection includes blackened color of the burned area, evolution of partial necrosis to total necrosis, greenish coloration of the subcutaneous tissue, appearance of vesicles in healed lesions, quick detachment of the necrotic tissue, and the presence of phlogistic signs (hyperemia and edema) in areas close to burns [4].

Microbiota of healthy intact skin is characterized by the presence of some microorganisms, e.g., *Staphylococcus epidermidis*, *Staphylococcus aureus*, *Streptococcus sp*., *Escherichia coli* (perineum), *Pseudomonas aeruginosa* (underarms and inguinal regions) and *Candida albicans* [5]. If not immune-compromised, these microorganisms do not represent an issue to intact skin, but they could represent a health problem in burned skin.

Severe burns are commonly treated with silver sulfadiazine 1% cream, which is applied onto the burned area, followed by protection with bandages composed of several layers of gauze and cotton. These dressings require frequent replacement because the bactericidal action of the cream lasts a maximum of 12 h. The cream may even dry over time, with the consequent adherence of the dressing onto the wound surface, leading to pain, emotional trauma and considerable damage to the newly formed epithelium when the dressing is again removed [6,7]. Another disadvantage regarding the use of the gauze dressings is the risk of almost occlusion of the wound and the accumulation of fluid below the dressing, which favors the proliferation of pathogens and delays the healing process [7,8].

In this perspective, biopolymeric dressings, like those based on chitosan films, are promising to overcome these limitations. Moreover, chitosan exhibits antibacterial and antifungal activity [9], and also acts as an agent that assists in the natural blood coagulation, which serves as a protection for the nerves endings, reducing the pain [10].

The chemical structure of chitosan is similar to the structure of hyaluronic acid, which reinforces the indication of the use of this biopolymer as a healing agent. Besides, chitosan is capable of enhancing the role of inflammatory cells (e.g., polymorphonuclear leukocytes, macrophages) in promoting the cellular organization and in acting in the repair of large wounds [11]. Due to these properties, one of the most extensive medical applications for chitosan is the production of films to be used as burn wound dressing, hemostatic agent, and material for surgical suture [12].

Zeolites are aluminosilicates widely used in the chemical industry. The high surface of zeolites can be exploited for the incorporation of molecules, like AgSD, for sustained release devices. The sustained release can even be enhanced by the inclusion of these zeolites into chitosan films [13]. The AgSD-impregnated zeolite can deliver AgSD directly to the wound and at a proper rate to act against microorganisms and promote fast healing. The tetrahedral structure of zeolite are arranged in rings, which are combined to form regular and uniform channels and cavities in which Ag is included and the release occurs over time [14].

The novelty of our work grounds on the development of an ideal dressing that combines the chitosan properties with the AgSD antimicrobial properties, together with the sustained release of sulfadiazine assisted by the zeolite. Chitosan film with silver sulfadiazine (zeolite free) was also prepared and evaluated for comparison. The resulting materials were physicochemically characterized, evaluated their antimicrobial properties and cytotoxicity profile in Balb/c 3T3 fibroblasts.

## 2. Materials and Methods

### 2.1. Materials

Silver sulfadiazine (AgSD) was obtained from J.P.N. Phamar PVT Ltd. (Mumbai, India). NaY zeolite was used with a particle size of 0.054 mm (Fábrica Carioca de Catalisadores, Rio de Janeiro, Brazil). Ammonium hydroxide was supplied by Sigma-Aldrich (São Paulo, Brazil). Commercial chitosan (deacetylation of approximately 82% and molar mass of about 1.47 × 10^5^ g/mol) was supplied by Polymar (Fortaleza, Brazil) and used for the film preparation without prior purification. For antimicrobial tests, strains of *Escherichia coli* (ATCC 8739), *Staphylococcus aureus* (ATCC 6538), *Pseudomonas aeruginosa* (ATCC 9027) and *Candida albicans* (ATCC 10231) were used. Tryptic Soy Broth-Soybean-Casein Digest Medium (Bacto^TM^) and Sabouraud Dextrose Broth (Acumedia, Indaiatuba, SP, Brazil) were used as culture broth to grow the microorganisms.

### 2.2. Impregnation of Silver Sulfadiazine

Wet impregnation was used to add AgSD to the zeolite, applying the methodology adapted from Yassue-Cordeiro et al. [14]. The AgSD/zeolite ratio was defined as 5 g of AgSD/100 g of the total sample. Zeolite was firstly dried in a Q316M oven (Quimis, São Paulo, Brazil) at 100 °C for 24 h. Silver sulfadiazine was solubilized in ammonium hydroxide solution (30% *w/v*), followed by the evaporation of the solvent at 70 °C under vacuum. The resulting material was dried at 100 °C for 12 h in the same oven.

### 2.3. Preparation of Chitosan Films

Chitosan was firstly solubilized in an aqueous solution containing acetic acid added in stoichiometric amount, plus 50% excess based on the chitosan mass and the deacetylation degree. The solution was kept under magnetic stirring for 2 h for complete solubilization.

Silver sulfadiazine-impregnated zeolite and glycerol (plasticizer) were added to chitosan solution in the following ratio: 0.2 g of AgSD-zeolite/100 g of chitosan solution, and 25 g of glycerol/100 g of dry chitosan powder. The amount of zeolite was based on preliminary work [15], whereas the amount of glycerol was consistent with that defined by Thakhiew et al. [16]. The suspension was homogenized under mechanical stirring for 2 h at 2000 rpm. Then, it was transferred to a Büchner flask for air removal under a vacuum pump for 10 to 15 min. The suspension was then poured into polyethylene Petri dishes (12.5 cm × 12.5 cm), which were subsequently dried in an oven with forced air circulation at 37 °C for 24 h (TE-394-4 model, Tecnal, Brazil). A constant ratio of 0.21 g of suspension/cm² of the dish was used. After drying, films were carefully removed from the dishes and properly stored in plastic containers to protect against direct light exposure and excessive moisture. Films with AgSD (zeolite free) were also prepared in the same way for comparison.

### 2.4. Characterization of Materials

#### 2.4.1. Nuclear Magnetic Resonance of ^29^Si and ^27^Al

The solid-state NMR experiments were carried out at 59.6132 MHz for the nucleus frequency of ^29^Si and 78.186 MHz for ^27^Al, equipped with solid probe CP/MAS 7 mm (Model 300, Varian Mercury Plus, Palo Alto, CA, USA). MAS technique was used for readings of both silicon and aluminum, using the following parameters: acquisition time 0.050 s, a pulse of 90° and room temperature. The silicon samples were placed in a 7 mm-diameter dimension zirconia rotor with Kel-F and applying the rotational speed of 3.5 kHz. The aluminum samples were put in a 7 mm-diameter dimension silicon nitride rotor with Kel-F cover and applying the speed rotational of 6.5 kHz. The spectrum of kaolin (signal at −91.16 ppm) was used as the external reference.

#### 2.4.2. X-ray Fluorescence by Total Reflection (TXRF)

The X-ray fluorescence by total reflection (TXRF) analysis was used to determine the amount of silver in the zeolite after the impregnation. Samples were previously prepared and fixed in quartz reflectors, which were irradiated for 800 s in total reflection by a beam of X-rays of 20 keV, extracted from the radiation source of molybdenum, for determining the concentration of elements present in the samples. From the disk, a TXRF spectrum was generated, and the peak intensities of the lines K, L and M processed regarding elemental concentrations (mg kg^−1^) by own equipment software (S2 PICOFOX, Billerica, MA, USA).

#### 2.4.3. Scanning Electron Microscopy (SEM)

The micrographs of the zeolites samples were obtained using a scanning electron microscope (Shimadzu SS-550, Kyoto, Japan). In the same microscope, the analysis of the surface chemical composition of zeolites with AgSD was also performed using an energy-dispersive X-ray spectrometer probe (EDX). This analysis allows mapping of the principal components to check the dispersion of these elements on the surface of samples. The morphological aspect of the zeolite-chitosan composite films was examined by scanning electron microscopy (440i LEO Electron Microscopy Ltd., Cambridge, UK) with 10 kV and 100 pcA. For the SEM analysis, the sample was previously coated with a thin gold layer (~10 nm) of a conductive material following the analysis of the cryogenically fractured cross-sections.

#### 2.4.4. X-ray Diffraction Spectroscopy

XRD graphs were obtained with an X-ray diffractometer (XRD, Diffraktometer Bruker D8 Advance, Billerica, MA, USA) for zeolite and film samples. The X-ray diffractometer was operated at 40 kV and 50 mA, with Cu-Kα radiation (λ = 1.54056 Å). Samples were scanned from 4° to 50° (2θ) at a scanning rate of 0.24°/min and a step of 0.02°. All the XRD patterns were interpreted using the JCPDS database software.

#### 2.4.5. Thermal Analysis

Differential scanning calorimetry (DSC) and termogravimetric analysis (TGA) studies were performed on zeolite-chitosan composite films. TGA was done with a TGA-60 (Shimadzu, Kyoto, Japan) analyzer. All analyses were performed with 10~15 mg samples in platinum pans in a dynamic nitrogen atmosphere (100 mL/min), between 30 °C and 700 °C (10 °C/min). DSC analysis was performed with a DSC-60 (Shimadzu, Kyoto, Japan) analyzer. Samples (approximately 2–3.5 mg) were scanned in a sealed aluminum pan and heated to 550 °C (10 °C/min) under the nitrogen atmosphere (50 mL/min). The weight loss (TGA) and the enthalpies (DSC) were calculated using the TA.60 software provided by Shimadzu.

#### 2.4.6. Fourier Transform Infrared Spectroscopy

The spectra of powder zeolite were obtained using a spectrophotometer (Bruker Vertex 70v, Billerica, MA, USA). The reading was performed in the medium infrared region with Fourier transform (FTIR) in 400 a 4000 cm^−1^ and a resolution of 4 cm^−1^. Samples were pressed with potassium bromide (KBr) powder to obtain a 0.5% pellet sample.

The polymeric films were analyzed with the spectrometer IRPrestige21 (Shimadzu, Kyoto, Japan) using the Attenuated Full Reflection (ATR) accessory. The ATR device allows getting information about the surface chemical structure. The non-normalized spectra (15 scans) of chitosan films were recorded, and the samples were analyzed between 650 and 4000 cm^–1^ with a resolution of 4 cm^–1^.

#### 2.4.7. Fluid Handling Capacity (FHC)

The fluid handling capacity (FHC) of the film is defined as the sum of the Absorbency (ABS) and the Moisture Vapor Transmission Rate (MVTR). The FHC was examined, as stated by the BS EN 13726-1 [17] method for hydrocolloids and dressings. In this test, samples were mounted into the modified Paddington cups, to which a volume of 20 mL of simulated exudate fluid (SEF) was added. SEF is an isotonic sodium/calcium chloride solution containing 142 mM sodium ions and 2.5 mM calcium ions [18]. These concentrations of cations represent the salt concentration observed in the fluid wound and serum.

The cups were weighted using a calibrated analytical balance; cups were placed in an inverted position so that the dressing could come into contact with the SEF. The devices were placed in a controlled incubator at 37 °C ± 2 and relative humidity below 20% for 24 h. At the end of the test, the cups were removed from the incubator and were allowed to equilibrate at room temperature for 30 min before reweighting on the analytical balance. The FHC, ABS, and MVTR were calculated using the following equations:(1)$$MVTR=\frac{x-y}{times\times surface}$$
(2)$$ABS=\frac{b-a}{time\times surface}$$
(3)FHC=MVTR+ABS
where *x* is the weight of the complete system (film + SEF solution + cup) at the beginning of the test; *y* is the weight of the complete system (film + SEF solution + cup) after 24 h; *b* is the weight of the film at the beginning of the test; and *a* is the weight of the film after 24 h. Five repetitions were done per experiment.

#### 2.4.8. Mechanical Properties

Tensile testing was done in agreement with the ASTM D882 method [19]. Films were cut into 10.00 cm × 2.54 cm strips. The tensile strength, elongation at breaking point and Young’s modulus were measured using TexturePro CT V1.2 (Brookfield, CT3 50K Texturometer, Middleboro, MA, USA). The crosshead speed was set at 1 mm/s. Samples were pre-conditioned in a desiccator at 75% relative humidity at equilibrium for 48 h. At least 10 repetitions per experiment were performed.

#### 2.4.9. Silver Sulfadiazine Release Test

To evaluate the release profile of AgSD from chitosan films, a simulated exudate fluid (SEF, 142 mmol/L sodium ions and 2.5 mmol/L calcium ions) representing the salt concentrations observed in wound fluids was used as a medium and serum at a controlled temperature of 37 °C. Release tests were performed using 4 × 4 cm film samples (average mass 0.15 g). Erlenmeyer flasks with the film samples and 100 mL of SEF solution were placed on an orbital shaker (Fisherbrand™, Fisher Scientific, Waltham, MA, USA) and kept under constant stirring throughout the assay to decrease the risk of mass convection resistance. Aliquots of 1.5 mL were taken at predetermined times over a total test time of 48 h. The analysis was performed in duplicate and the silver released was analyzed by atomic absorption spectroscopy (AA SpectrAA 50B, Varian, Palo Alto, CA, USA) using a hollow cathode lamp (λ = 328 nm) and a mixture of air and acetylene generated the flame.

#### 2.4.10. Antimicrobial Activity of Chitosan Films

Antimicrobial activity analyses against common human pathogens, namely, *Escherichia coli* (ATCC 8739), *Staphylococcus aureus* (ATCC 6538), *Pseudomonas aeruginosa* (ATCC 9027) and *Candida albicans* (ATCC 10231), were carried out with the AgSD-loaded films. The strains were standardized using the technique of series dilution. Analyses were carried out by the inoculation of 10^6^ CFU/mL of each strain in individual tubes containing 30 mL of TSB medium for the bacteria or 30 mL SDB medium for the yeast. The samples of films were cut into dimensions of 6 × 6 cm, and the sterilization were performed with UV light for 15 min each side. All sterilized samples were incubated in sterile test tubes. For the positive control, each tube was incubated with the microorganism only, whereas for the negative control, the test tubes contained only TSB or SDB medium. The tubes were incubated at 37 °C for 12 h. Aliquots of 100 µL were transferred to flat-bottomed microplates. The microbiological growth was monitored by reading the optical density by an automatic microplate reader (Synergy HT Biotek, Winooski, VT, USA) at a wavelength of 620 nm.

#### 2.4.11. In Vitro Indirect Cytotoxicity

Balb/c 3T3 fibroblasts were grown in DMEM (Vitrocell Embriolife) supplemented with 10% fetal bovine serum (Vitrocell Embriolife), as recommended by ISO 10993 [20]. When fibroblasts reached 80% confluency, they were detached from the bottles by the action of trypsin (0.05% trypsin solution and 0.02% EDTA in phosphate buffer pH 7.2) and plated in 96-well plates (P96w) with 20,000 cells per well. Each sample film was cut and sterilized by exposure to a UV lamp for 15 min on both sides. After sterilization, the films were placed separately in sterile tubes containing serum (6 cm^2^/mL) for 72 h at 37 °C to prepare the extracts. Extracts were filtered with membranes of 0.45 µm (nominal pore diameter), and seven dilutions were carried out. Each dilution was performed in sextuplicate. The medium was removed from the 96-well plate containing the fibroblasts, and the diluted extracts were added to fibroblasts cells. The plates were placed in a humid incubator (5% CO_2_ and 37 °C) for 24 h. Then, extracts were removed, the wells washed twice with phosphate buffer pH 7.2 and a new medium with Neutral red was added for color development. The absorbance was measured in a microplate reader at a wavelength of 540 nm. The cell viability was calculated using the Equation (4).
(4)$$CV(\%)=\left(\frac{OD_{sample}}{OD_{control}}\right)x100$$
where CV is the cell viability, *OD_sample_* is the optical density of the extracted sample, *OD_control_* is the optical density of the test control cells.

#### 2.4.12. Statistical Analysis

One-way analysis of variance (ANOVA) and *T*-test were employed for statistical analysis. *P* ≤ 0.05 was considered indicative of a statistically significant difference. The study was conducted using the software Microcal Origin v. 7.0 (Origin Lab Corp, Wellesley Hills, MA, USA).

## 3. Results and Discussion

### 3.1. Characterization of Silver Zeolite

The ^29^Si MAS-NMR de-convoluted spectrum of NaY zeolite revealed the presence of Si(3Al), Si(2Al), Si(1Al) and Si(0Al) ambient, relative to −89.27; −93.77; −99.10; and −104.61 ppm, respectively, as described in the literature [21,22]. The Si/Al molar ratio of the structure was about 2.54. As reported by Weitkamp and Puppe [23], the Si/Al ratio of the NaY zeolite ranges from 1.5 to 3. Concerning the ^27^Al MAS-NMR spectrum, a resonance line at 59.02 ppm corresponding to the signal of tetrahedral aluminum species was identified. The absence of peak related to octahedral aluminum, a signal approximately at 0 ppm, indicated the absence of extra-framework aluminum, which was also described by Guerra et al. [24].

The chemical composition of the zeolite with silver sulfadiazine sample was carried out by TXRF and EDX. The silver content of zeolite was 2.26%, which is in agreement with the TXRF analysis. On the other hand, EDX spectrum exhibited the presence of oxygen (44.60% *w/w*), sodium (7.35%), aluminum (9.14% *w/w*), silicon (24.53% *w/w*) and silver (3.55%), but also gold and carbon (4.01% *w/w*) peaks due to the metallization of the samples and the double-sided tape used for fixing the sample on the support, respectively. The emission of the first ten of the lower atomic numbers consists of bands in the low energy region where the losses by absorption in the sample are more significant, resulting in no detection or no precision of these elements [25]. The obtained silver content of 3.55% mass was expected (less than 5%) since 5% corresponds to the AgSD and not only to silver. The Si/Al ratio determined by EDX was 2.58, which is very close to the NMR results.

The divergence between the results recorded with TXRF and EDX was attributed to the capacity of TXRF to quantify the total silver bulk mass, whereas EDX quantifies silver up to a depth of 500 nm [25]. As reported by Boschetto et al. [26], EDX identifies the silver on the surface of the zeolite. Using ICP-OES (Inductively Coupled Plasma Optical Emission Spectroscopy) analysis, the authors observed that the real silver content approached the theoretical content while the amount of silver quantified by EDX was two-fold higher than the theoretical value, attributed to the higher Ag^+^ concentration on the external surface [26]. Important to mention is that ICP-OES analysis also provides the amount of bulk silver.

Figure 1 illustrates the mapping component of the silver sulfadiazine-impregnated zeolite analyzed by EDX for aluminum (b), silicon (c), oxygen (d), sodium (e), and silver (f). The silver is dispersed over the whole surface of the zeolite due to the large size of the AgSD molecules. These AgSD crystals outside the zeolite structure are long and needle-like (Figure 1a and Figure 2), and they consist mostly of silver atoms. Scanning electron microscopy (SEM) analysis (Figure 2) was performed to obtain high-resolution image of the surface of the powder zeolite after impregnation.

Silver sulfadiazine is a large molecule, and its morphology is modified when impregnated in the zeolitic support. Small “needles” of silver sulfadiazine with dimensions of 33.75 µm in length and 7.14 µm in width were observed.

Zeolite samples were also analyzed by X-ray diffraction (XRD) by the powder method. The XRD pattern shown in Figure 3 was analyzed with the assistance of the Bank of JCPDS data. The characteristic peaks of FAU zeolite depicted in 2θ = 6.24°, 15.70°, 23.69°, 27.08° and 31.43° assigned, respectively, to the crystallographic planes [111], [331], [622], [624], [804] have already been observed in the zeolite NaY [14]. It would have been expected to detect the characteristic peaks of metallic silver at 38.21°, 44.51°, 64.51, and 77.51° in XRD diffractograms of the AgSD-Y (Figure 3). Some authors have even used several methodologies to incorporate silver in zeolite structures, which were then analyzed by XRD and no silver crystals could also be identified [26,27,28,29,30,31,32,33]. The fact is, not only AgSD exhibits a strong reflection at 2θ = 10.2°, but zeolite Y shows the very same peak [34].

Figure 4a shows the differential scanning calorimetry (DSC) curves of silver sulfadiazine powder. According to Bult and Plug [34], the DSC results of silver sulfadiazine are very much dependent on the atmosphere where the study is conducted. In our work, the analysis was performed under helium or nitrogen atmosphere resulting in an endothermic peak between 283 °C and 300 °C, whereas the exothermic peak was recorded at about 290 °C. Such exothermic event is not found in sulfadiazine or sodium sulfadiazine, and is therefore related to a chemical reaction of silver or catalyzed by silver. Moreover, the endothermic process overlaps with the melting range of silver sulfadiazine. Both peaks are shown in Figure 4a, i.e., an endothermic peak at 299.20 °C, which is related to the melting event with an onset temperature at 293 °C, and an exothermic peak at 296.88 °C which is associated to the decomposition of silver sulfadiazine.

The DSC curves shown in Figure 4b follow the pattern of zeolite Y. The endothermic peak (first peak) corresponds to the loss of moisture in the zeolite or physiosorbed water [35]. One endothermic shoulder was observed for Y zeolite at around 256–257 °C, which has been attributed to the dehydration of hydrated sodium ions located in the zeolite structure [36]. Silver sulfadiazine shifted these two peaks to a slightly higher temperature (257.81 °C). AgSD-Y presented a third peak, i.e., an exothermic peak at 477.24 °C, which was associated with the decomposition of silver sulfadiazine. In this case, zeolite would be promoting the protection of AgSD from the decomposition that usually occurs at around 300 °C.

Thermogravimetric analyses (TGA) for silver sulfadiazine powder and zeolites are illustrated in Figure 5. Silver sulfadiazine exhibited a higher weight loss (~51%) between 250 °C and 350 °C, corresponding to its decomposition. Both zeolites (NaY and AgSD-Y) showed significant weight loss between 50 °C and 100 °C. The AgSD-Y sample had a higher total cumulative weight loss (17.16%) compared to the NaY zeolite (15.17%) up to 700 °C. The incorporation of silver sulfadiazine in the zeolitic support improved the hygroscopic character of AgSD-Y, resulting in increased water retention and lower stability.

Table 1 and Figure 6 illustrate the main peaks obtained by FTIR analysis for the vibrations found in NaY zeolite and the AgSD-Y zeolite. All bands corresponding to lattice vibrations reported by Flanigen et al. [37] in the spectral region of 1300–400 cm^−1^ were identified in our samples after impregnation with AgSD. According to Nadtochenko et al. [38] and to Mohseni-Bandpi et al. [39], the band around 1634 cm^−1^ is due to bending vibration of OH from adsorbed water. No significant displacements or enlargements in the characteristic bands are exhibited in the spectra (Figure 6a), meaning that no decrease of crystallinity occurred when compared to the NaY zeolite, which corroborates the results obtained by XRD [14]. However, this sample shows additional peaks that can also be identified in the spectrum of silver sulfadiazine powder (Figure 6b), occurring at 1552, 1500, 1420 and 1356 cm^−1^ regarding the S=O bound. This result suggests that the sulfadiazine molecule is adsorbed onto the zeolite surface [34,40]. On the other hand, Bult and Plug [34] suggested that infrared peaks at 1500, 1560 and 1595 cm^−1^ recorded for silver sulfadiazine correspond to phenyl skeletal vibration, pyrimidine skeletal vibration, and phenyl skeletal vibration, respectively. Some peaks between 1500 and 1600 cm^−1^ were observed for AgSD-Y, which could be associated with the presence of AgSD in the films.

### 3.2. Characterization of the Polymer Films

The previously characterized zeolites were added to the chitosan films and, for comparison purposes, films without AgSD-Y and films with pure AgSD were also prepared. Chitosan films with or without zeolites were characterized for their physicochemical, mechanical and barrier properties. The thicknesses of pure chitosan films and chitosan films with pure AgSD obtained were 0.040–0.055 mm and in films with zeolite 0.060–0.070 mm. SEM micrographs of the surface of the polymer films (magnification of 500×) and the cross-section (magnification of 3000×) are shown in Figure 7.

Pure chitosan films were of light-yellow transparent color, flexible and with a reasonable handling resistance. Macroscopic and microscopic evaluation showed uniform surface without the presence of defects [14], as small holes, fissures or macropores, which was also reported by other authors [41,42,43]. The small amount of pure AgSD added to the chitosan film was calculated by the silver sulfadiazine present in the AgSD-Y. Both films had the same amount of AgSD, differing only in the presence of zeolite. Silver sulfadiazine did not change the color if compared to the pure chitosan films. Moreover, the micrographs (Figure 7a,b) revealed no flaws or AgSD agglomerates. On the other hand, the addition of zeolites promoted an increase in the opacity (close to gray color) and rigidity of the films. The AgSD-Y-loaded films presented large clusters of zeolite, as observed in cross-sectional micrographs (Figure 7d). We can also notice the presence of small cracks in the micrograph of the cross-section (along the agglomeration), which can be attributed to possible physical change or interaction between the zeolite particles and the polymer chains detected in XRD diffractograms.

X-ray diffraction analyses were conducted to obtain information about the structures of chitosan powder, and pure chitosan film (Figure 8a). Furthermore, Figure 8b allows comparing the diffractograms of all the films and zeolite AgSD-Y as well.

According to Wani et al. [44], chitosan powder can be identified in three structural forms: hydrate crystal, anhydrous crystal and non-crystal or amorphous. The hydrate crystal has high diffraction at 2θ = 10° and other less pronounced peaks at 20° and 22°. The anhydrous crystal exhibits a high peak at 2θ = 15° and an additional peak at 20°. In the amorphous phase, no peak was observed any, but a broad halo exists at 2θ = 20°. In the chitosan used in this study (Figure 8a), the presence of hydrate and anhydrous crystals could be observed, as well as the strong presence of amorphous phase, characteristic of semi-crystalline polymers [45,46,47,48]. Peaks at 26.83° and 29.45° are also typical of powder chitosan reported in the literature [43].

The semi-crystalline profile of the powder chitosan is due to strong intra- and intermolecular interactions, characterized by hydrogen bonds formed between amine, alcohol, and amide groups or other functional groups, providing particular organization to the crystalline structure of the polysaccharide [49]. The intensity of the peak at 20° is drastically reduced (57.39%) when chitosan recrystallized during the drying process of the film, which has also been reported by other authors [44,50]. The incorporation of zeolite into the polymer matrix of the film (Figure 8b) promotes characteristic peaks of zeolite in the XRD patterns of the film. The addition of zeolite particles interferes with the orderly align of chitosan chains both by steric effects and by the formation of hydrogen bonds between the –OH surface groups of zeolites and –NH_2_ e –OH of chitosan. Thus, all the samples presented a decrease in crystallinity of the chitosan and the appearance of characteristic peaks of zeolite [51,52,53].

The heating curves showed the presence of three main peaks in all produced films (Figure 9), which corroborates previous findings [50,54,55]. The first one is an endothermic event attributed to water evaporation linked to the chitosan chain through physical bonds (hydrogen bonds, electrostatic interactions) [50,54,55]. It was recorded at 66.75, 65.53 and 65.14 °C, for chitosan film, AgSD-Y/Chitosan film and AgSD/Chitosan film, respectively. The second one is an exothermic event, observed at 213.06, 213.85 and 210.88 °C, respectively, for the chitosan film, the AgSD-Y/Chitosan film and the AgSD/Chitosan film. The third one is also an exothermic peak at 275.56, 283.86 and 278.65 °C, respectively, for the chitosan film, the AgSD-Y/Chitosan film and the AgSD/Chitosan film. Both the second and third peaks are associated with the decomposition of chitosan [56]. The addition of Ag-Clinoptilolite zeolites in chitosan films also increased the decomposition temperature of the films and decreased the evaporation temperature of water. Figure 9 shows that the incorporation of zeolite increases the heat of water evaporation enthalpy. The hydrogen bonds between hydroxyl groups of chitosan and the Si–O–Si groups of zeolite would compromise the interaction of the hydroxyl groups of chitosan with the water [13].

Films exhibited three main stages of mass loss, namely, 30–150 °C, 180–240 °C, and 240–420 °C (Figure 9). The first stage was attributed to the evaporation of adsorbed water, the second stage to the degradation of the chitosan chain (during which other chemical species are obtained) and the third to the decomposition of residual organic groups (during which changes in the physicochemical properties of chitosan occur) [42,50,55,57]. The third stage had the highest mass loss of all samples. Besides the degradation and decomposition of chitosan, this stage can also occur when glycerol starts degrading in the range of 180–289 °C [58].

The TGA results show that after heating up to 700°, the mass loss was 66.29%, 67.90%, and 71.48%, for AgSD-Y/Chitosan film (blue line), chitosan film (black line), and AgSD/Chitosan film (red line), respectively (Figure 10). The AgSD/chitosan film showed the highest mass loss because AgSD enhances the hydrophilicity of the film, with consequently higher water retention in the polymeric matrix, resulting in lower thermal stability of the film. The AgSD-Y/Chitosan film also showed more significant weight loss compared to the pure chitosan film even because AgSD increased the hydrophilicity of the film. Moreover, the zeolite would not suffer decomposition anyway, thus the final mass was expected to be higher. The highest mass loss was observed in the third mass loss stage; in this stage, the lowest intensity DTG peak of AgSD-Y/Chitosan film indicated lower decomposition velocity, which may be an indication that the zeolite decreases the decomposition rate of the chitosan film. The DTG peak height at any temperature gives the mass change ratio at that temperature.

Figure 11 illustrates the FTIR spectra obtained for the pure chitosan films, the films with pure AgSD and the films with incorporated zeolites. Table 2 shows the peaks obtained for all films along with their respective assignments. The addition of pure silver sulfadiazine to the films did not result in any change in the infra-red spectra compared to those of pure chitosan. The incorporation of zeolite particles in polymeric films resulted in some changes in the spectra compared to that of pure chitosan film. These changes in the intensity of the bands of the hydroxyl group, amide I and amide II of the films may have been caused by hydrogen bonds between the –OH groups of zeolites with –OH and –NH_2_ groups of chitosan [42,52,53]. Moreover, AgSD-Y-loaded chitosan films exhibited just one band from the stretching C–O frequencies at 1017 cm^−1^, which may be either the combination of the bands at 1061 cm^−1^ and 1015 cm^−1^ or the overlapping of the band Si-O of zeolites with the stretching band –C–O of chitosan.

Table 3 depicts the moisture vapor transmission rate (MVTR), absorbency (ABS), fluid handling capacity (FHC) values and mechanical properties of chitosan films. The MVTR values for the films ranged from 2.52 ± 1.03 to 3.04 ± 1.49 g/10 cm^2^/24 h; however, no significant differences were observed between them. These results are also consistent with those reported for ActivHeal^®^ and Allevyn Adhesive^®^ commercial dressings, which exhibited MVTR of 1.67g ± 0.11 g/10 cm^2^/24 h and 12.35 ± 0.42 g/10 cm^2^/24 h, respectively [18]_._ Other commercial dressings, as Allevyn Gentlee Mepilex^®^, Tielle Plus^®^, Allevyn Adhesive Border^®^, Allevyn Gentle^®^, Biatain Adhesive^®^, also depicted MVTR values in this range [59,60,61].

Water loss by evaporation of skins that have suffered first-degree burns is 0.279 ± 0.026 g/cm²/24 h and the wound granulation stage is 5.138 ± 0.202 g/cm²/24 h [62]. Thus, for a dressing to be used in the treatment of burns, its permeability rates should be between 2.0 and 2.5 g/cm²/24 h to promote a proper humidity level without the risk of wound bed dehydrating [62,63]. Other authors suggest that MVTR must be higher than the exudate production, ranging between 3.4 and 5.1 g of exudate/10 cm^2^/24 h [18,64]. The MVTR for the synthesized films showed intermediate values, allowing to confirm that our chitosan films could be used as a wound dressing.

The significant difference between AgSD-Y/Chitosan film and other films is due to their lower thermal stability, attributed to the high hydrophilicity of silver sulfadiazine powder. Consequently, there was greater water retention in the polymeric matrix resulting in more top ABS and FHC if we compare to other synthesized films. As described by Thomas and Young [18], the rate of production of exudate varies during the healing process of a wound; in other words, it is higher at the beginning and decreases until eventually ceases when the wound closes. Thus, the use of AgSD-Y-loaded chitosan film is suitable for the treatment just after wound formation, while the other films could be used in the final stage when high absorption capacity is not so necessary.

Table 3 also shows the mechanical properties obtained for the chitosan films. Concerning the AgSD-Y/Chitosan film, we observed an increase in tensile strength and a decrease in the percentage of elongation. Similar results were found by Vicentini et al. [52] and Cui et al. [65], who suggested the presence of zeolites, promoting electrostatic interactions between the polymers and zeolites, restricting the mobility of the polymer chains.

According to Dallan [58], the presence of chitin clusters caused local disorganization in the packaging of chitosan polymeric chains making the polymer structure more easily ruptured near the points where the chitin clusters are inserted, reducing the elongation percentage of the chitosan films with chitin when compared to films composed only of chitosan. These results are consistent with those observed in AgSD-Y/Chitosan film, which exhibited many zeolite clusters on the surface of the film, fact observed by SEM, which may have caused the low percentage of elongation in our samples. As shown by SEM analysis of cross-section, AgSD-Y-loaded chitosan film exhibited agglomerates (Figure 7). The decrease in the elongation at break (Table 3) could be attributed to the presence of these agglomerates which facilitates the breaking. For the AgSD-loaded chitosan films, the decreasing could be attributed to the discontinuities of the chitosan chain produced by the crystallization of AgSD during the drying. The presence of glycerol in polymeric films resulted in films with enhanced flexibility (lower Young’s modulus) and improved flexibility (more significant elongation percentage) if compared to the films prepared by Santos et al. [61] in which ZSM-5 was added to the chitosan film without the presence of glycerol.

Figure 12 shows the release profiles of silver ions from chitosan films. There was no difference between the silver release from AgSD-Y/Chitosan film and AgSD/Chitosan film up to 1500 min of the test. After this time-length, the silver released from AgSD-Y/Chitosan film increased, while AgSD/Chitosan film did not change, meaning that zeolites enhance the sustained release of silver.

The AgSD/Chitosan film exhibited a saturated release profile (Figure 12). The maximum amount of silver released from this sample was about 0.5 ppm (or 10% of the initial amount of silver added to the chitosan film). Although the small percentage, it is the amount that could be available for the release; the remaining silver is entrapped in the interstices of the polymer chains or within the zeolite cavities. Statistically significant differences were seen between both profiles. AgSD-Y/Chitosan film did not reach the saturation, even by the end of the assay (3000 min). As discussed above, the incorporation of silver sulfadiazine in the zeolitic support increases the hygroscopic character of AgSD-Y, resulting in increased water retention; the released silver increases, which is highly desirable since silver can be cytotoxic. Two days after starting the release tests, silver was still being released from the films, and the concentration was less than 1 ppm.

Walker et al. [66] found that Ancticoat^®^ and Aquacel-Ag Hydrofiber^®^ commercial dressings reached about 55 and 1 ppm of Ag^+^, respectively. Aquacel-Ag Hydrofiber^®^ released such a low amount of silver as the films synthesized in this work. However, the low silver concentration obtained was also reported by other authors, who performed the experiment in a saline solution like simulated exudate fluid and observed that concentrations of the ion silver were about 1 ppm in all cases [66,67].

This analysis was decided to be performed in the simulated solution of wound exudate since this is closer to the real physiological conditions found in a burn—although the chloride ions in this solution may partially inhibit the silver ions. Although not solely being performed in deionized water, and not resembling the real in vivo situation [68,69], no considerable amount of silver was detected. There was also no ion exchange to promote the release of silver from the zeolite structure.

### 3.3. Antimicrobial Activity of the Polymer Films

Figure 13 shows the inhibitory tests of the chitosan films against *P. aeruginosa*, *E. coli*, *S. aureus,* and *C. albicans*. All the evaluated microorganisms were not significantly inhibited by pure chitosan films. Inhibitory activity of chitosan films against *E. coli, S. aureus, S. typhimurium, L. monocytogenes* and *B. cereus* were also not reported by previous works [70,71,72]. The active agents, or the protonated soluble chitosan fraction, do not migrate to the medium and the biocide effect was not observed. The antimicrobial activity of insoluble chitosan films is low because the interaction of protonated groups of chitosan with negative groups of microbial cell walls was reported to be weak [73].

Chitosan presented excellent biocide properties only in the form of gels or solutions because the amino groups of the biopolymer are almost entirely protonated under these conditions [72]. López-Caballero et al. [74] also investigated the antimicrobial effect of chitosan on the growth of gram-negative bacteria and found that the addition of chitosan in bacterial culture resulted in no inhibition of microbial growth, which was attributed to the reduced solubility of chitosan in the neutral pH and the presence of non-charged amino groups [75].

There are other factors that must be considered in the antimicrobial properties of chitosan, such as the molar mass (10,000 to 100,000 Da), the degree of deacetylation, the ionizable amino groups, acidic solvents, the availability of lipids and proteins as interference, pH, pKa, surface charge, the concentration of chitosan, the ionic strength of the matrix, reaction time, the chelate ability and types of bacteria [76]. Both films with AgSD (65% reduction of cell viability) were active against the proliferation of *C. albicans* and had a lower activity against bacterial strains (*P. aeruginosa*, *E. coli*, *S. aureus*). Fajardo et al. [8] synthesized dressings based on silver sulfadiazine/chitosan/chondroitin sulfate and found high antimicrobial activity against *P. aeruginosa* and *S. aureus*. In our samples, the amount of AgSD added was relatively low, thereby requiring a more significant amount to achieve a higher antimicrobial activity.

Concerning the films with pure AgSD and AgSD-Y, gram-negative strains (*P. aeruginosa* and *E. coli*) were more susceptible to the action of silver than gram-positive bacteria (*S. aureus*); and this was attributed to differences in the composition of the cell walls of these bacteria. Gram-positive bacteria have a higher resistance against Ag^+^ ions because their cell walls contain a larger amount of peptidoglycan (3–20 times), providing an additional barrier against antimicrobial agents [28,69,77].

*Candida albicans* was sensitive to the action of all the synthesized films. These results were also reported by Ferreira et al. [31], and it was attributed to the complex cellular organization of eukaryotic cells and/or cell wall structure. Silver attacks the plasmatic membrane of the yeast, resulting in the formation of pores that interrupts the membrane potential, inactivates the process of budding and subsequently occurs cell death.

### 3.4. In Vitro Indirect Cytotoxicity

Figure 14 shows the effect of concentration of the extract obtained from the films on the inhibition of fibroblast growth. The extract containing AgSD-zeolite showed a higher capacity to inhibit the growth of fibroblasts. For this film, up to the extract concentration of 25%, the cell viability was just over 60%, considering the standard deviation, these values can be at best close to 70%, which would be the minimum acceptable value for viability cell so that the material is not considered cytotoxic [20]. For the extract containing AgSD, the cell viability was above 70% except for undiluted extract (concentration of 100%). Therefore, both extracts without dilution (concentration of 100%) inhibited the growth of fibroblasts at levels that are considered cytotoxic.

Abe et al. [78] studied the incorporation of silver zeolite in tissue conditioners, which are used in dental prostheses. They observed that the higher the zeolite content, the greater the inhibition of human fibroblast growth. The cytotoxic effect was checked for four of the five samples of tissue conditioners, and the cell viabilities of these samples were less than 40%. Except for one sample, cell viability was not modified by the addition of the zeolite, indicating the importance of the composition of the material in cytotoxicity. This work shows the importance of cytotoxicity of zeolites.

Our films containing zeolite exhibited higher cytotoxicity than films without them. Although the cytotoxicity of silver sulfadiazine has been much reported in the literature, cytotoxicity data for zeolite are almost non-existent. Important to mention that there is a wide variety of zeolites, natural and synthetic, and for the same type of zeolite, the composition thereof may vary significantly. Gorinova et al. [79] assessed in vitro the cytotoxicity of mesoporous silica MCM-41 toward human hepatocellular carcinoma cells HepG2. They tested MCM-41 empty or loaded with sulfadiazine (not silver sulfadiazine). Conforming to the results, MCM-41 showed no cytotoxicity at concentrations up to 0.2 mg/mL; however, it increases above 1 mg/mL. This work could help us to explain the cytotoxicity of films containing AgSD-Y zeolite, as zeolites are the kind of material very similar to mesoporous silica.

Duc et al. [80] evaluated the potential cytotoxic effects of some antiseptics on human skin substitutes compared to autograft skin. They found that a commercial ointment of silver sulfadiazine and one local brand ointment (silver sulfadiazine 1% with 1% acetic acid) showed, respectively, moderate and high cytotoxicity against dermal fibroblasts and keratinocytes besides a detrimental effect on tissue histology when applied to autograft and especially to human skin substitutes. Moreover, they concluded that antiseptics that were classified as safe to use in the in vitro study are usually safe to use in vivo. Antiseptics that were classified as cytotoxic in their in vitro study may even be slightly less cytotoxic in vivo, mainly if a high amount of wound exudate is present. Thus, although our films have shown some cytotoxicity, it is likely that the in vivo test results are promising.

Gao et al. [81] assessed the cytotoxic effect of AgSD bulk powder and AgSD nanocrystals incorporated in genipin crosslinked chitosan hydrogel. They found that the encapsulation of nanocrystals significantly reduced cytotoxicity. On the other hand, AgSD nanocrystals at the same concentration presented less cell viability compared to AgSD bulk powders. The high surface-to-volume ratio of nanocrystal increased interaction between fibroblasts and particles.

## 4. Conclusions

The polymeric chitosan films with dispersed AgSD or AgSD-impregnated zeolite demonstrated to be potential materials for the development of a new dressing. The impregnation process modifies the morphology of the starting zeolite. The addition of AgSD into the films did not result in substantial changes in the infra-red spectra, the XRD diffraction patterns, and the micrographs when compared to the pure chitosan films. In contrast, the addition of AgSD-Y zeolite resulted in less transparent films, with agglomerations of zeolites on the film structure. In these films, the presence of small agglomerates observed in the micrograph of the cross-section could be attributed to the dispersion of the zeolite powder into the chitosan solution and also to a possible physical interaction between the zeolite particles and the polymer chains detected in XRD diffractograms and infrared spectra. Moreover, the use of zeolites promoted a sustained release of the silver ions. The synthesized films showed some antimicrobial activity against the proliferation of *C. albicans* and a lower activity against bacterial strains. Gram-negative bacteria (*P. aeruginosa* and *E.coli*) were more susceptible to the action of silver than gram-positive bacteria (*S. aureus*) due to compositional differences in cell walls. The safety evaluation studies showed that although the AgSD-zeolite films exhibited a cytotoxicity profile at the tested conditions of 72 h of extraction, it is possible to adjust the user conditions and, in time, the concentration dependent fashion so that the films would not pose any toxicological risk.

## Figures and Tables

**Figure 1 pharmaceutics-11-00535-f001:**
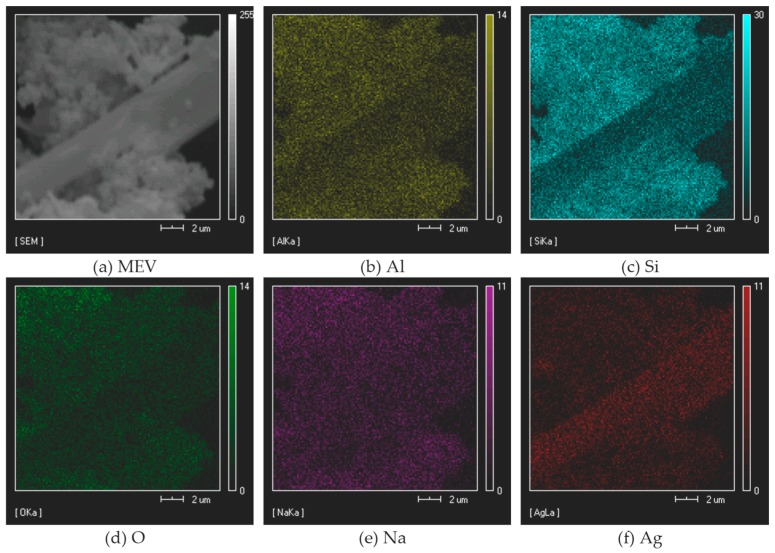
Component performed by X-ray spectrometer probe (EDX) for the zeolite AgSD-Y. (**a**) original image; (**b**) Al; (**c**) Si; (**d**) O; (**e**) Na; (**f**) Ag.

**Figure 2 pharmaceutics-11-00535-f002:**
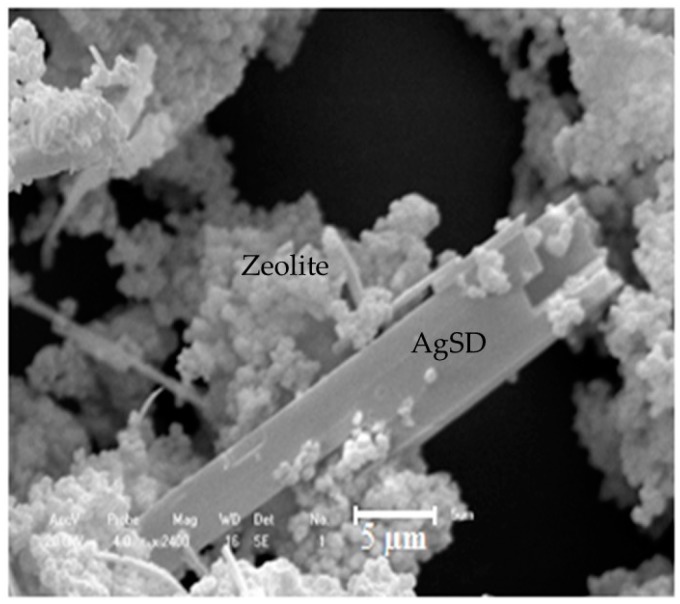
SEM micrograph of AgSD-Y zeolite (magnification of 2400×).

**Figure 3 pharmaceutics-11-00535-f003:**
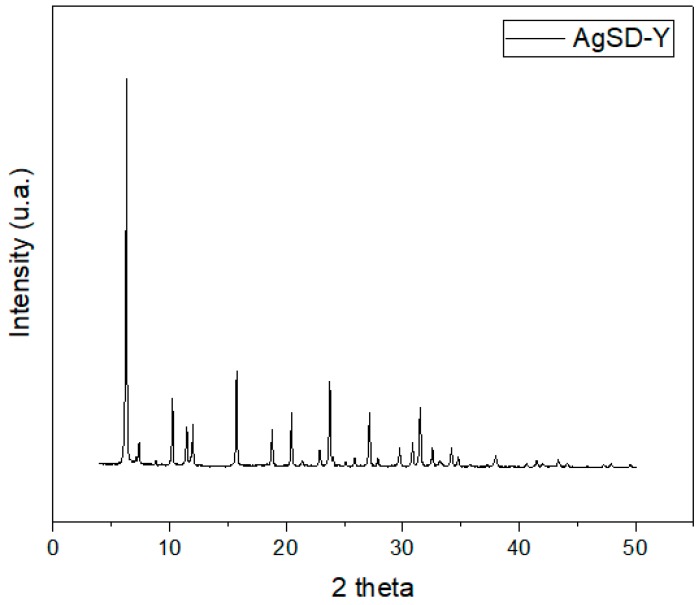
Diffractograms of AgSD-Y zeolite.

**Figure 4 pharmaceutics-11-00535-f004:**
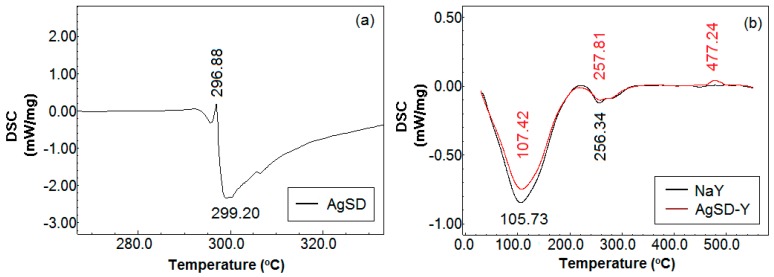
Heating curve of differential scanning calorimetry (DSC) of (**a**) pure AgSD and (**b**) NaY and AgSD-Y zeolite.

**Figure 5 pharmaceutics-11-00535-f005:**
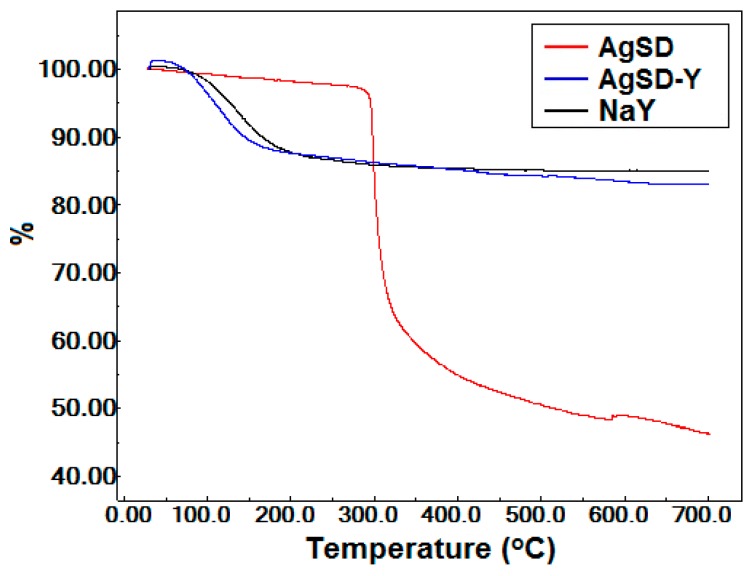
Curve of AgSD, AgSD-Y and NaY samples.

**Figure 6 pharmaceutics-11-00535-f006:**
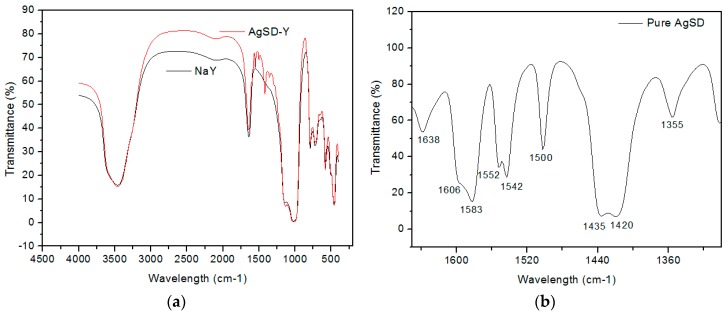
Spectrum of (**a**) NaY, AgSD-Y zeolite and (**b**) pure AgSD.

**Figure 7 pharmaceutics-11-00535-f007:**
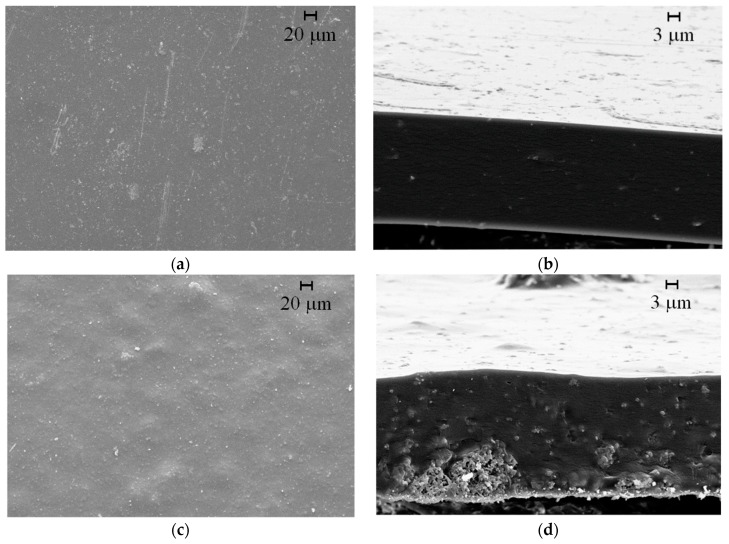
SEM micrographs of chitosan film (left: surface at magnification 500×); right: cross-section at magnification 3000×): AgSD/chitosan film (**a,b**); AgSD-Y/Chitosan film (**c,d**).

**Figure 8 pharmaceutics-11-00535-f008:**
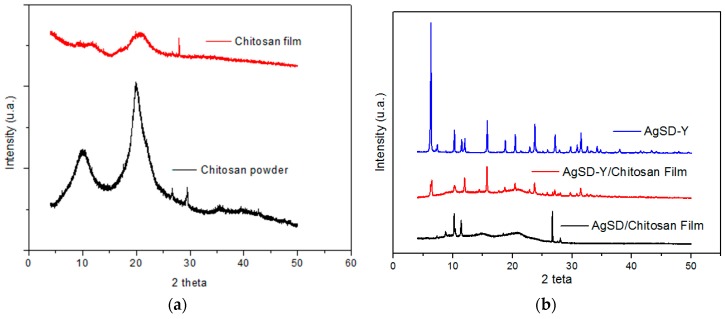
Diffractograms of (**a**) chitosan films, chitosan powder and (**b**) AgSD-Y powder, AgSD-Y/chitosan film and AgSD/chitosan film.

**Figure 9 pharmaceutics-11-00535-f009:**
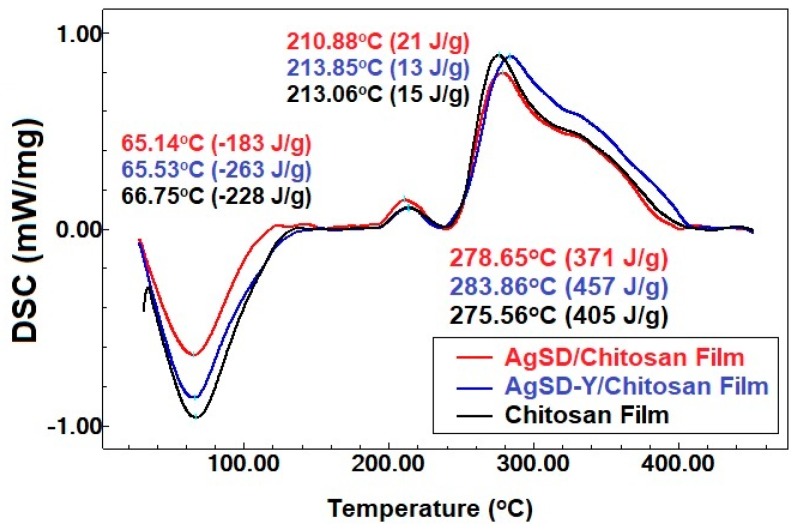
Heating curves of Chitosan films, AgSD/Chitosan film, and AgSD-Y/Chitosan film.

**Figure 10 pharmaceutics-11-00535-f010:**
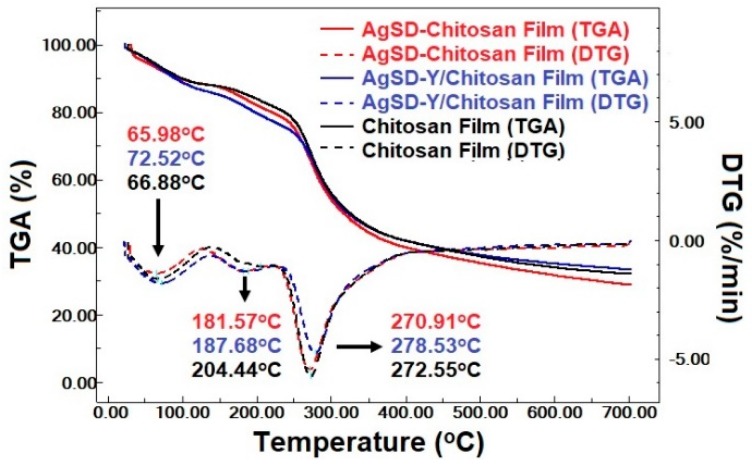
Curve for the Chitosan film and AgSD/Chitosan film and AgSD-Y/Chitosan film.

**Figure 11 pharmaceutics-11-00535-f011:**
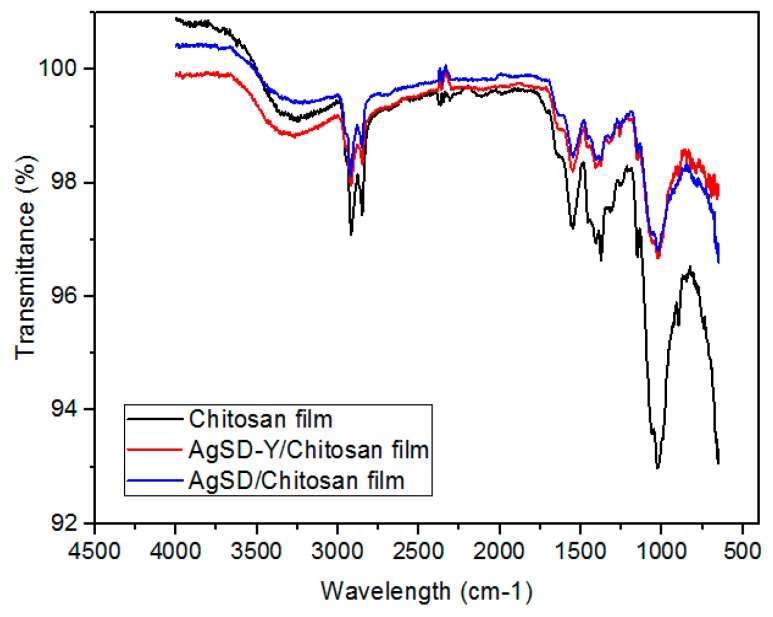
Infrared spectrum of the chitosan film, AgSD-Y/Chitosan film and AgSD/Chitosan film.

**Figure 12 pharmaceutics-11-00535-f012:**
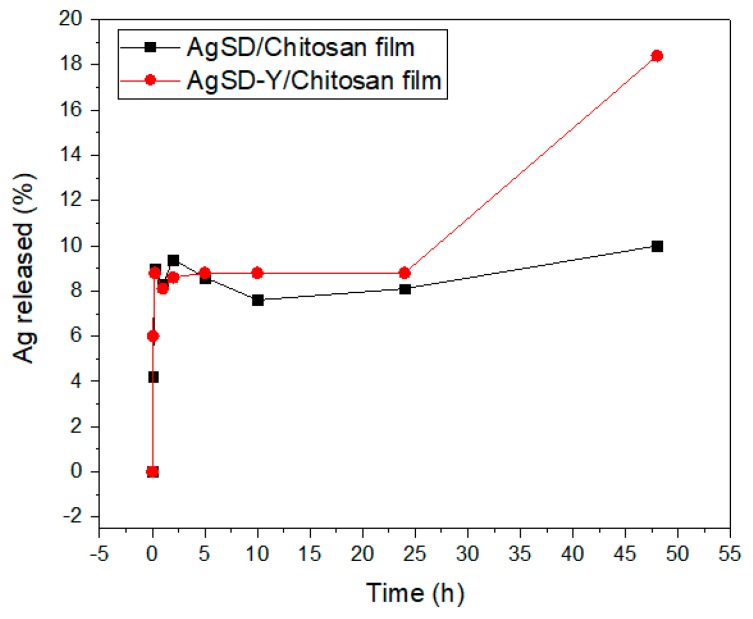
Release-test from AgSD/Chitosan film and AgSD-Y/Chitosan film.

**Figure 13 pharmaceutics-11-00535-f013:**
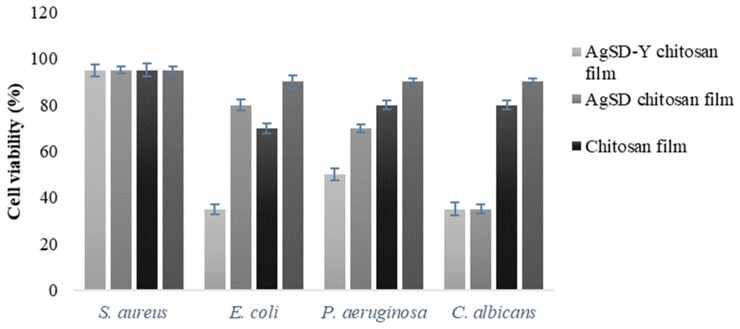
Tests (cell viability of *S. aureous*, *E. coli, P. aeruginosa* and *C. albicans*) in liquid medium containing AgSD-Y/Chitosan film, AgSD/Chitosan film, Chitosan film or AgNO_3_. Results are the mean ± standard deviation (*n* = 3).

**Figure 14 pharmaceutics-11-00535-f014:**
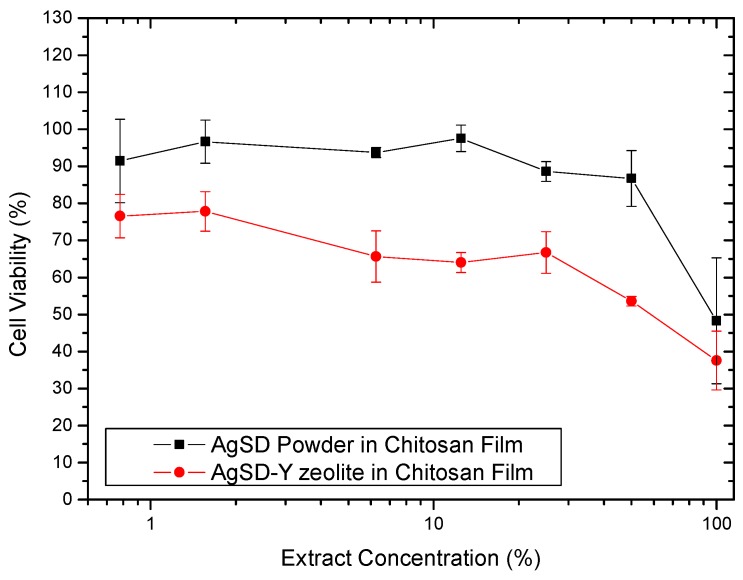
Effect of extract concentration on growth inhibition of Balb/c 3T3 fibroblasts with AgSD/Chitosan film and AgSD-Y/Chitosan film.

**Table 1 pharmaceutics-11-00535-t001:** Vibrational bands obtained for NaY and AgSD-Y zeolite.

	Assignment	Band (cm^−1^)	
		NaY	AgSD-Y
	T-OH surface	3470	3470
	Zeolitic water	1643	1650
Internal	Stretching asymmetric	1143	1140
	Stretching symmetric	720	720
	Angular deformation	501/461	502/460
External	Double rings(D6R)s	577	580
	Stretching symmetric	791	792
	Stretching asymmetric	1026	1027

**Table 2 pharmaceutics-11-00535-t002:** Vibrational bands obtained for Chitosan films, AgSD/Chitosan film and AgSD-Y/Chitosan film.

Assignment	Band (cm^−1^)		
	Films		
	Chitosan	AgSD/Chitosan	AgSD-Y/Chitosan
Stretching O-H and N–H	3320	3320	3319
Stretching C–H	2919/2845	2919/2845	2919/2845
Amide I	1650	1650	1648
Amide II	1550	1550	1564
Stretching C–H e O–H	1411	1411	1411
Angular deformation C–H	1365	1365	1365
Stretching C–O–C	1155	1155	1156
Stretching C–O	1015/1061	1015/1061	1017
Presence –C–H.	893	893	890

**Table 3 pharmaceutics-11-00535-t003:** Moisture Vapor Transmission Rate (MVTR), Absorbency (ABS), Fluid Handling Capacity (FHC), and mechanical properties of chitosan film, AgSD/Chitosan film and AgSD-Y/Chitosan film (adapted from [14]).

Films	MVTR	ABS	FHC	Young’s Modulus (MPa)	Tensile Strength (MPa)	Elongation at Break (%)
Chitosan	2.52 ± 1.03 ^a^	1.57 ± 0.44 ^a^	4.09 ± 1.47	4.17 ± 0.97 ^a^	13.26 ± 0.71 ^a^	19.60 ± 0.38 ^a^
AgSD-Y	2.52 ± 0.22 ^a^	4.25 ± 1.15 ^b^	7.29 ± 2.23	5.08 ± 1.55 ^a^	15.81 ± 1.12 ^b^	12.74 ± 3.03 ^b^
AgSD	3.04 ± 1.49 ^a^	2.34 ± 0.67 ^a^	5.15 ± 0.87	6.85 ± 1.91 ^b^	15.16 ± 2.71 ^a^	8.71 ± 1.97 ^b^

Note: Different superscripts in the same column and line indicate significant differences between formulations (*p* < 0.05); MVTR, ABS, and FHC are expressed in g/10 cm^2^ 24 h.

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
