# Peer review of "Development of Chitosan/Silver Sulfadiazine/Zeolite Composite Films for Wound Dressing"

_pharmaceutics, 2019, doi:10.3390/pharmaceutics11100535_

Round 1
Reviewer 1 Report
Review of the manuscript: pharmaceutics-536690
Title: Development of chitosan/silver sulfadiazine/zeolite composite films for wound dressing.
The objective of the experimental work presented in this manuscript is to develop and study silver sulfadiazine (AgSD) impregnated zeolites incorporated into chitosan films as a potential skin wound dressing. The zeolite would allow a controlled and sustained release of the (AgSD), controlling skin infection and reducing cytotoxicity of AgSD particles.
The experimental work included the preparation of the AgSD-zeolite (AgSD-Y) and its characterisation by several techniques; solid-state NMR, TXRF, XRD, SEM, FTIR.
AgSD-Y-chitosan prepared films were characterised by FTIR, tensile tests, DSC, TGA and transport properties using NaCl solutions. The latter was complemented AgSD particles release studies from the films, antimicrobial properties and cytotoxicity assessment using 3T3 fibroblasts.
The work contributes with valuable information about zeolites based systems for controlled release of bioactive particles. Also in terms of the prepared zeolite and chitosan composite film structure at molecular and macroscopic levels. However, the manuscript suffers some problems that need to be addressed prior to publication if Pharmaceutics.
First, the novelty of the work should be highlighted in the introduction, results and conclusion sections.
The presentation and discussion of the results referred to zeolite and film characterization should be directed towards the controlled release of AgSD as indicated in the introduction. In the current version, this section seems as separated work from the antimicrobial activity and cytotoxicity work. It needs a more integrated discussion.
•The numerous acronyms in the text that need to be defined more clearly. Also, their use should be consistent throughout the text.
•In the Material and Methods section, each technique section should be presented in the same order as in the sections in the Results and Discussion.
•The text in the figures’ captions throughout the manuscript should be completed and be self-explanatory.
•Lines 75-76. Provide references for this statement.
•Lines 84-85. Why the AgSD zeolites can improve their sustain release when incorporated to chitosan film.
•Lines 163.164. Provide more information about sample coating for SEM measurements.
•Provide information on how statistical differences were calculated.
•Section 2.4.5. Was FTIR spectra data normalised?
•Figure 1 is not required for the manuscript and should be removed. Too many figures already.
•Section 2.4.7. Provide info about the final moisture content of the films equilibrated under 75% RH.
•Section 2.4.8. Were the samples in powdered or film form? Why delta H is not considered in the discussions?
•Section 2.4.9. Sample weight? Was AgSD in zeolites? Or powder AgSD in chitosan films?
•Sections 2.4.10 and 2.4.11 Was AgSD in zeolites? Or powder AgSD in chitosan films? Is the effect of UV light on AgSD negligible?
•Quality of images in figure 2 should be improved. It is hard to see differences apart from the differences in colour.
•In figure 3, please mark AgSD and the zeolites in the picture.
•Picture quality if figure 4 should be improved. Also, the peak at 10o is not discussed.
•Figure 6. Consider including d%/dT curves to help identify differences in drying kinetics between samples.
•Lines 314 and 315. Check the writing, is confusing as it is.
•Table 1. Provide evidence for zeolitic water identified at band 1634-1650 cm-1
•Line 382. Sorry but I don’t see the cracks.
•Lines 409-410. Crystallinity calculations are required to support this statement.
•Line 419. Add references to support this statement.
•Lines 425-427. Explain the differences between chitosan degradation and decomposition. Why both transitions were exothermic?
•Line 430. Are you referring to 200C? At 700C AgSD-Y chitosan sample lost more weight.
•In Table 3, all samples analysed were chitosan films? Some samples are referred to as AgSD and AgSD-Y. Were the moisture and glycerol the same in all films?
• Add statistics in figure 13.
Author Response
Reviewer #1
The objective of the experimental work presented in this manuscript is to develop and study silver sulfadiazine (AgSD) impregnated zeolites incorporated into chitosan films as a potential skin wound dressing. The zeolite would allow a controlled and sustained release of the (AgSD), controlling skin infection and reducing cytotoxicity of AgSD particles.
The experimental work included the preparation of the AgSD-zeolite (AgSD-Y) and its characterisation by several techniques; solid-state NMR, TXRF, XRD, SEM, FTIR.
AgSD-Y-chitosan prepared films were characterised by FTIR, tensile tests, DSC, TGA and transport properties using NaCl solutions. The latter was complemented AgSD particles release studies from the films, antimicrobial properties and cytotoxicity assessment using 3T3 fibroblasts.
The work contributes with valuable information about zeolites based systems for controlled release of bioactive particles. Also in terms of the prepared zeolite and chitosan composite film structure at molecular and macroscopic levels. However, the manuscript suffers some problems that need to be addressed prior to publication if Pharmaceutics.
We acknowledge the referee’s comments and have addressed them carefully. The manuscript is annotated to make sure that all changes are clearly identified.
First, the novelty of the work should be highlighted in the introduction, results and conclusion sections.
We acknowledge the referee’s remark and have provided the missing information in the suggested sections.
In introduction, we have rephrased the last paragraph as follows: “The novelty of our work grounds on the development of an ideal dressing that combines the chitosan properties with the AgSD antimicrobial properties, together with the sustained release of sulfadiazine assisted by the zeolite. Chitosan film with silver sulfadiazine (zeolite free) was also prepared and evaluated for comparison. The resulting materials were physicochemically characterized, evaluated their antimicrobial properties and cytotoxicity profile in Balb/c 3T3 fibroblasts.”
In Results, it is evident the complete and large number of analytical techniques for characterization of the crystalline solid, that is, the zeolite (drug carrier) and the zeolite-loaded films, besides the biological characterization in vitro in the same article.
In Conclusion, although incorporation of the drug into zeolite did not improve cytotoxicity, it is possible to adjust time of use and concentration, but sustained release was substantially improved.
The presentation and discussion of the results referred to zeolite and film characterization should be directed towards the controlled release of AgSD as indicated in the introduction. In the current version, this section seems as separated work from the antimicrobial activity and cytotoxicity work. It needs a more integrated discussion.
We have thoroughly revised the manuscript and a more integrated discussion provided. The text is now given as follows: “The AgSD/Chitosan film exhibited a saturated release profile (Figure 13). The maximum amount of silver released from this sample was about 0.5 ppm (or 10% of the initial amount of silver added to the chitosan film). Although the small percentage, it is the amount that could be available for the release; the remaining silver is entrapped in the interstices of the polymer chains or within the zeolite cavities. Statistically significant differences were seen between both profiles. AgSD-Y/Chitosan film did not reach the saturation, even by the end of the assay (3000 min). As discussed above, the incorporation of silver sulfadiazine in the zeolitic support increases the hygroscopic character of AgSD-Y, resulting in increased water retention; the released silver increases, which is highly desirable since silver can be cytotoxic. Two days after starting the release tests, silver was still being released from the films, and the concentration was less than 1 ppm.”
The numerous acronyms in the text that need to be defined more clearly. Also, their use should be consistent throughout the text.
We have revised the abbreviations and acronyms. Changes are marked in red.
In the Material and Methods section, each technique section should be presented in the same order as in the sections in the Results and Discussion.
Section “Materials and Methods” has been revised and subsections renumbered accordingly. Section “Results and Discussion” provides the data in the same order as the previous section. A new subsection related to the statistical analysis included.
The text in the figures’ captions throughout the manuscript should be completed and be self-explanatory.
All legends of figures and tables have been carefully revised to be self-explanatory, as suggested. Edition is marked in red.
Lines 75-76. Provide references for this statement.
We acknlwegde the remark and the following references have been introduced accordingly:
[Reference 9] Mehrabani, M.G., et al., Chitin/silk fibroin/TiO2 bio-nanocomposite as a biocompatible wound dressing bandage with strong antimicrobial activity. International journal of biological macromolecules, 2018. 116: p. 966-976.
[Reference 10] Sanandiya, N.D., et al., Tunichrome-inspired pyrogallol functionalized chitosan for tissue adhesion and hemostasis. Carbohydrate polymers, 2019. 208: p. 77-85.
Lines 84-85. Why the AgSD zeolites can improve their sustain release when incorporated to chitosan film.
We acknlwegde the referee’s remark and justification has been provided in introduction as follows: “Zeolites are aluminosilicates widely used in the chemical industry. The high surface of zeolites can be exploited for the incorporation of molecules, like AgSD, for sustained release devices. The sustained release can even be enhanced by the inclusion of these zeolites into chitosan films [13]. The AgSD-impregnated zeolite can deliver AgSD directly to the wound and at a proper rate to act against microorganisms and promote fast healing. Tetrahedral structure of zeolite are arranged in rings, which are combined to form regular and uniform channels and cavities in which Ag is included and the release occurs over time [14].”, and a new reference introduced: “Yassue-Cordeiro, P.H., et al., Development and characterization of chitosan/silver zeolites composite films. Polímeros, 2015. 25(5): p. 492-502.”
Lines 163.164. Provide more information about sample coating for SEM measurements.
We acknlwegde the referee’s remark and the detailed technique has been provided in section 2.4.3. as follows: “The micrographs of the zeolites samples were obtained using a scanning electron microscope (Shimadzu SS-550, Japan). In the same microscope, the analysis of the surface chemical composition of zeolites with AgSD was also performed using an energy-dispersive X-ray spectrometer probe (EDX). This analysis allows mapping of the principal components to check the dispersion of these elements on the surface of samples. The morphological aspect of the zeolite-chitosan composite films was examined by scanning electron microscopy (440i LEO Electron Microscopy Ltda, England) with 10 kV and 100 pcA. For the SEM analysis, the sample was previously coated with a thin gold layer (~10 nm) of a conductive material following the analysis of the cryogenically fractured cross-sections.”.
Provide information on how statistical differences were calculated.
We acknlwegde the referee’s remark and the missing section (2.4.12) has been included as follows: “One-way analysis of variance (ANOVA) and T-test were employed for statistical analysis. P ≤ 0.05 was considered indicative of a statistically significant difference. The study was conducted using the software Microcal Origin v. 7.0 (Origin Lab Corp).”.
Section 2.4.5. Was FTIR spectra data normalised?
FTIR data have not been normalised. To clarify this query, this information has been given in section 2.4.6. as follows: “The non-normalized spectra (15 scans) of chitosan films were recorded, and the samples were analyzed between 650 and 4000 cm–1 with a resolution of 4 cm–1.”.
Figure 1 is not required for the manuscript and should be removed. Too many figures already.
We acknlwegde the referee’s remark and figure 1 has been removed as suggested.
Section 2.4.7. Provide info about the final moisture content of the films equilibrated under 75% RH.
The final moisture contents of the films were not measured. According to the literature, fhe films are usually kept at equilibrium under controlled relative humidity before proceeding to the analysis without assessed their moisture contents. To address this query, we have included the following information in section, now, 2.4.8.: “Tensile testing was done in agreement with the ASTM D882 method [19]. Films were cut into 10.00 cm × 2.54 cm strips. The tensile strength, elongation at breaking point and Young’s modulus were measured using TexturePro CT V1.2 (Brookfield, CT3 50K Texturometer, USA). The crosshead speed was set at 1 mm/s. Samples were pre-conditioned in a desiccator at 75% relative humidity at equilibrium for 48 h. At least 10 repetitions per experiment were performed.”.
Section 2.4.8. Were the samples in powdered or film form? Why delta H is not considered in the discussions?
The samples that have been analysed were in film form as stated in the section, now, 2.4.5. as follows: “Differential scanning calorimetry (DSC) and termogravimetric analysis (TGA) studies were performed on zeolite-chitosan composite films. TGA was done with a TGA-60 (Shimadzu, Japan) analyzer. All analyses were performed with 10 ~ 15 mg samples in platinum pans in a dynamic nitrogen atmosphere (100 mL/min), between 30°C and 700°C (10°C/min). DSC analysis was performed with a DSC-60 (Shimadzu, Japan) analyzer. Samples (approximately 2-3.5 mg) were scanned in a sealed aluminum pan and heated to 550°C (10°C/min) under the nitrogen atmosphere (50 mL/min). The weight loss (TGA) and the enthalpies (DSC) were calculated using the TA.60 software provided by Shimadzu.”. Thank your for your valuable contribution. The delta H was inserted in the DSC graph and discussed as well. The text has been edited as follows: “The heating curves showed the presence of three main peaks in all produced films (Figure 10), which corroborates previous findings [50, 54, 55]. The first one is an endothermic event attributed to water evaporation linked to the chitosan chain through physical bonds (hydrogen bonds, electrostatic interactions) [50, 54, 55]. It was recorded at 66.75, 65.53 and 65.14°C, for chitosan film, AgSD-Y/Chitosan film and AgSD/Chitosan film, respectively. The second one is an exothermic event, observed at 213.06, 213.85 and 210.88oC, respectively, for the chitosan film, the AgSD-Y/Chitosan film and the AgSD/Chitosan film. The third one is also an exothermic peak at 275.56, 283.86 and 278.65oC, respectively, for the chitosan film, the AgSD-Y/Chitosan film and the AgSD/Chitosan film. Both the second and third peaks are associated with the decomposition of chitosan [56]. The addition of Ag-Clinoptilolite zeolites in chitosan films also increased the decomposition temperature of the films and decreased the evaporation temperature of water. Figure 10 shows that the incorporation of zeolite increases the heat of water evaporation enthalpy. The hydrogen bonds between hydroxyl groups of chitosan and the Si–O–Si groups of zeolite would compromise the interaction of the hydroxyl groups of chitosan with the water [13].” and “The TGA results show that after heating up to 700°, the mass loss was 66.29%, 67.90%, and 71.48%, for AgSD-Y/Chitosan film (blue line), chitosan film (black line), and AgSD/Chitosan film (red line), respectively (Figure 11). The AgSD/chitosan film showed the highest mass loss because AgSD enhances the hydrophilicity of the film, with consequently higher water retention in the polymeric matrix, resulting in lower thermal stability of the film. The AgSD-Y/Chitosan film also showed more significant weight loss compared to the pure chitosan film even because AgSD increased the hydrophilicity of the film. Moreover, the zeolite would not suffer decomposition anyway, thus the final mass was expected to be higher. The highest mass loss was observed in the third mass loss stage; in this stage, the lowest intensity DTG peak of AgSD-Y/Chitosan film indicated lower decomposition velocity, which may be an indication that the zeolite decreases the decomposition rate of the chitosan film. The DTG peak height at any temperature gives the mass change ratio at that temperature.”.
Section 2.4.9. Sample weight? Was AgSD in zeolites? Or powder AgSD in chitosan films?
The results of the release profile of drugs from drug delivery devices are usually depicted and discussed in percentage. We have produced two distinct films, namely, AgSD/film chitosan and AgSD-Y/film chitosan. To clarify this section, we have provided it in more detail as follows: “To evaluate the release profile of AgSD from chitosan films, a simulated exudate fluid (SEF - 142 mmol/L sodium ions and 2.5 mmol/L calcium ions) representing the salt concentrations observed in wound fluids was used as medium and serum at a controlled temperature of 37°C. Release tests were performed using 4 x 4 cm film samples (average mass 0.15 g). Erlenmeyer flasks with the film samples and 100 mL of SEF solution were placed on an orbital shaker (Fisherbrand™, Fisher Scientific, Waltham, MA, USA) and kept under constant stirring throughout the assay to decrease the risk of mass convection resistance. Aliquots of 1.5 mL were taken at predetermined times over a total test time of 48 hours. The analysis was performed in duplicate and the silver released was analyzed by atomic absorption spectroscopy (AA SpectrAA 50B-Varian, USA) using a hollow cathode lamp (λ = 328 nm) and a mixture of air and acetylene generated the flame.”.
Sections 2.4.10 and 2.4.11 Was AgSD in zeolites? Or powder AgSD in chitosan films? Is the effect of UV light on AgSD negligible?
In both assays, AgSD in chitosan films were used. Also, the UV light is a usual technique of sterilization of films before biological assays. This information is given in section “2.4.10. Antimicrobial activity of chitosan films” as follows: “Antimicrobial activity analyses against common human pathogens, namely, Escherichia coli (ATCC 8739), Staphylococcus aureus (ATCC 6538), Pseudomonas aeruginosa (ATCC 9027) and Candida albicans (ATCC 10231), were carried out with the AgSD-loaded films. The strains were standardized using the technique of series dilution. Analyses were carried out by the inoculation of 106 CFU/mL of each strain in individual tubes containing 30 mL of TSB medium for the bacteria or 30 mL SDB medium for the yeast. The samples of films were cut into dimensions of 6 x 6 cm, and the sterilization were performed with UV light for 15 min each side. All sterilized samples were incubated in sterile test tubes. For the positive control, each tube was incubated with the microorganism only, whereas for the negative control the test tubes contained only TSB or SDB medium. The tubes were incubated at 37 °C for 12 h. Aliquots of 100 µL were transferred to flat-bottomed microplates. The microbiological growth was monitored by reading the optical density by an automatic microplate reader (Synergy HT Biotek, USA) at a wavelength of 620 nm.”, and in section “2.4.11 In vitro indirect cytotoxicity” as follows: “Balb/c 3T3 fibroblasts were grown in DMEM (Vitrocell Embriolife) supplemented with 10% fetal bovine serum (Vitrocell Embriolife), as recommended by ISO 10993 [20]. When fibroblasts reached 80% confluency, they were detached from the bottles by the action of trypsin (0.05% trypsin solution and 0.02% EDTA in phosphate buffer pH 7.2) and plated in 96-well plates (P96w) with 20,000 cells per well. Each sample film was cut and sterilized by exposure to a UV lamp for 15 min on both sides. After sterilization, the films were placed separately in sterile tubes containing serum (6 cm2/mL) for 72 h at 37oC to prepare the extracts. Extracts were filtered with membranes of 0.45 µm (nominal pore diameter), and seven dilutions were carried out. Each dilution was performed in sextuplicate. The medium was removed from the 96-well plate containing the fibroblasts, and the diluted extracts were added to fibroblasts cells. The plates were placed in a humid incubator (5% CO2 and 37oC) for 24 h. Then, extracts were removed, the wells washed twice with phosphate buffer pH 7.2 and a new medium with Neutral red was added for color development. The absorbance was measured in a microplate reader at a wavelength of 540 nm. The cell viability was calculated using the equation (04).”.
Quality of images in figure 2 should be improved. It is hard to see differences apart from the differences in colour.
Figure 2 has been improved, as suggested. The figure with higher resolution has been provided as follows:
In figure 3, please mark AgSD and the zeolites in the picture.
Figure 3 has been improved, as suggested. The updated figure has been provided as follows:
Picture quality if figure 4 should be improved. Also, the peak at 10o is not discussed.
Figure 4 has been improved, as suggested. The figure with higher resolution has been provided as follows:
Figure 6. Consider including d%/dT curves to help identify differences in drying kinetics between samples.
We acknowledge the referee’s remark. The DTG curves have been provided in the revised version of the manuscript and discussed in the text.
Lines 314 and 315. Check the writing, is confusing as it is.
We acknowledge the referee’s remark. The text has been revised as follows: “Figure 5a shows the differential scanning calorimetry (DSC) curves of silver sulfadiazine powder. According to Bult and Plug [34], the DSC results of silver sulfadiazine are very much dependent on the atmosphere where the study is conducted. In our work, the analysis was performed under helium or nitrogen atmosphere resulting in an endothermic peak between 283ºC and 300oC, whereas the exothermic peak was recorded at about 290ºC. Such exothermic event is not found in sulfadiazine or sodium sulfadiazine, and is therefore related to a chemical reaction of silver or catalyzed by silver. Moreover, the endothermic process overlaps with the melting range of silver sulfadiazine. Both peaks are shown in Figure 5a, i.e. an endothermic peak at 299.20oC, which is related to the melting event with an onset temperature at 293oC, and an exothermic peak at 296.88oC which is associated to the decomposition of silver sulfadiazine.”
Table 1. Provide evidence for zeolitic water identified at band 1634-1650 cm-1
According to Nadtochenko et al. (2005) and to Mohseni-Bandpi et al. (2016) , the band around 1634 cm−1 is due to bending vibration of OH from adsorbed water. This information has been included in the manuscript as follows: “Table 1 and Figure 7 illustrate the main peaks obtained by FTIR analysis for the vibrations found in NaY zeolite and the AgSD-Y zeolite. All bands corresponding to lattice vibrations reported by Flanigen et al. [37] in the spectral region of 1300-400 cm-1 were identified in our samples after impregnation with AgSD. According to Nadtochenko et al. [38] and to Mohseni-Bandpi et al. [39], the band around 1634 cm−1 is due to bending vibration of OH from adsorbed water. No significant displacements or enlargements in the characteristic bands are exhibited in the spectra (Figure 7a), meaning that no decrease of crystallinity occurred when compared to the NaY zeolite, which corroborates the results obtained by XRD [14]. However, this sample shows additional peaks that can also be identified in the spectrum of silver sulfadiazine powder (Figure 7b), occurring at 1552, 1500, 1420 and 1356 cm-1 regarding the S=O bound. This result suggests that the sulfadiazine molecule is adsorbed onto the zeolite surface [34, 40]. On the other hand, Bult and Plug [34] suggested that infrared peaks at 1500, 1560 and 1595 cm-1 recorded for silver sulfadiazine correspond to phenyl skeletal vibration, pyrimidine skeletal vibration, and phenyl skeletal vibration, respectively. Some peaks between 1500 and 1600 cm-1 were observed for AgSD-Y, which could be associated with the presence of AgSD in the films.”. The following references have been introduced:
[38] Nadtochenko, V.A., et al., Dynamics of E. coli membrane cell peroxidation during TiO2 photocatalysis studied by ATR-FTIR spectroscopy and AFM microscopy. Journal of Photochemistry and Photobiology A: Chemistry, 2005. 169(2): p. 131-137.
[39] Mohseni-Bandpi, A., et al., Improvement of zeolite adsorption capacity for cephalexin by coating with magnetic Fe3O4 nanoparticles. Journal of Molecular Liquids, 2016. 218: p. 615-624.
Line 382. Sorry but I don’t see the cracks.
Indeed, a more careful analysis of Figure 8d (cross-section) allows the identification of small cracks along the agglomeration. To drive the reader to our conclusions, we have edited the paragraph as follows: “Pure chitosan films were of light-yellow transparent color, flexible and with a reasonable handling resistance. Macroscopic and microscopic evaluation showed uniform surface without the presence of defects [14], as small holes, fissures or macropores, which was also reported by other authors [41-43]. The small amount of pure AgSD added to the chitosan film was calculated by the silver sulfadiazine present in the AgSD-Y. Both films had the same amount of AgSD, differing only in the presence of zeolite. Silver sulfadiazine did not change the color if compared to the pure chitosan films. Moreover, the micrographs (Figure 8 a and b) revealed no flaws or AgSD agglomerates. On the other hand, the addition of zeolites promoted an increase in the opacity (close to gray color) and rigidity of the films. The AgSD-Y-loaded films presented large clusters of zeolite, as observed in cross-sectional micrographs (Figure 8d). We can also notice the presence of small cracks in the micrograph of the cross-section (along the agglomeration), which can be attributed to possible physical change or interaction between the zeolite particles and the polymer chains detected in XRD diffractograms.”.
Lines 409-410. Crystallinity calculations are required to support this statement.
The intensity of the peak at 20° was calculated and the reduction was 57.39%. The missing information has now been provided as follows: “The semi-crystalline profile of the powder chitosan is due to strong intra- and intermolecular interactions, characterized by hydrogen bonds formed between amine, alcohol, and amide groups or other functional groups, providing particular organization to the crystalline structure of the polysaccharide [49]. The intensity of the peak at 20° is drastically reduced (57,39%) when chitosan recrystallized during the drying process of the film, which has also been reported by other authors [44, 50]. The incorporation of zeolite into the polymer matrix of the film (Figure 9b) promotes characteristic peaks of zeolite in the XRD patterns of the film. The addition of zeolite particles interferes with the orderly align of chitosan chains both by steric effects and by the formation of hydrogen bonds between the –OH surface groups of zeolites and –NH2 e –OH of chitosan. Thus, all the samples presented a decrease in crystallinity of the chitosan and the appearance of characteristic peaks of zeolite [51-53]”.
Line 419. Add references to support this statement.
We acknowledge the referee’s remark and have included the reference “Qu, X., A. Wirsén, and A.C. Albertsson, Effect of lactic/glycolic acid side chains on the thermal degradation kinetics of chitosan derivatives. Polymer, 2000. 41(13): p. 4841-4847.” to support the statement, as follows: “The heating curves showed the presence of three main peaks in all produced films (Figure 10), which corroborates previous findings [50, 54, 55]. The first one is an endothermic event attributed to water evaporation linked to the chitosan chain through physical bonds (hydrogen bonds, electrostatic interactions) [50, 54, 55]. It was recorded at 66.75, 65.53 and 65.14°C, for chitosan film, AgSD-Y/Chitosan film and AgSD/Chitosan film, respectively. The second one is an exothermic event, observed at 213.06, 213.85 and 210.88oC, respectively, for the chitosan film, the AgSD-Y/Chitosan film and the AgSD/Chitosan film. The third one is also an exothermic peak at 275.56, 283.86 and 278.65oC, respectively, for the chitosan film, the AgSD-Y/Chitosan film and the AgSD/Chitosan film. Both the second and third peaks are associated with the decomposition of chitosan [56]. The addition of Ag-Clinoptilolite zeolites in chitosan films also increased the decomposition temperature of the films and decreased the evaporation temperature of water. Figure 10 shows that the incorporation of zeolite increases the heat of water evaporation enthalpy. The hydrogen bonds between hydroxyl groups of chitosan and the Si–O–Si groups of zeolite would compromise the interaction of the hydroxyl groups of chitosan with the water [13].”.
Lines 425-427. Explain the differences between chitosan degradation and decomposition. Why both transitions were exothermic?Thermal decomposition is a process involving changes in chemical species caused by heat whereas thermal degradation is a process whereby the action of heat or elevated temperature on material, product, or assembly causes a loss of physical, mechanical or electric properties. Both are exothermic as result in mass loss and release of energy. We acknowledge the referee’s remark and to clarify this query we have edited the paragraph as follows: “Films exhibited three main stages of mass loss, namely, 30-150°C, 180-240°C, and 240-420°C (Figure 10). The first stage was attributed to the evaporation of adsorbed water, the second stage to the degradation of the chitosan chain (during which other chemical species are obtained) and the third to the decomposition of residual organic groups (during which changes in the physicochemical properties of chitosan occur) [42, 50, 55, 57]. The third stage had the highest mass loss of all samples. Besides the degradation and decomposition of chitosan, this stage can also occur when glycerol starts degrading in the range of 180-289°C [58].”.
Line 430. Are you referring to 200C? At 700C AgSD-Y chitosan sample lost more weight.The statement “The third stage had the highest mass loss of all samples. Besides the degradation and decomposition of chitosan, this stage can also occur when glycerol starts degrading in the range of 180-289°C [58].” refers to all the samples as the results are very close among the films.
In Table 3, all samples analyzed were chitosan films? Some samples are referred to as AgSD and AgSD-Y. Were the moisture and glycerol the same in all films?As stated in Table 3: “Table 3. MVTR, ABS, FHC, and mechanical properties of chitosan film, AgSD/Chitosan film and AgSD-Y/Chitosan film (adapted from [14]).”, all samples were chitosan films, according to what is listed in the first column of the table.
Add statistics in figure 13.
The statistical analysis was done. We have included the missing information as follows: “The AgSD/Chitosan film exhibited a saturated release profile (Figure 13). The maximum amount of silver released from this sample was about 0.5 ppm (or 10% of the initial amount of silver added to the chitosan film). Although the small percentage, it is the amount that could be available for the release; the remaining silver is entrapped in the interstices of the polymer chains or within the zeolite cavities. Statistically significant differences were seen between both profiles. AgSD-Y/Chitosan film did not reach the saturation, even by the end of the assay (3000 min). As discussed above, the incorporation of silver sulfadiazine in the zeolitic support increases the hygroscopic character of AgSD-Y, resulting in increased water retention; the released silver increases, which is highly desirable since silver can be cytotoxic. Two days after starting the release tests, silver was still being released from the films, and the concentration was less than 1 ppm.”.

Reviewer 2 Report
This is an extensive study on the development and analysis of a AgSD containing film.
The introduction needs to be re-written and proper references are required. At present it is not accurate and parts are not relevant. Especially the background regarding the impact of burn wounds on patients and the description of infection are improper. Overall, grammar needs to be checked.
Gauze dressings are not occlusive but obstruct the wound from observation by clinicians.
A similar chemical structure does not not predict biological function.
Page 3 please define “properly stored”.
What is the added value of figure 1?
2.4.9 Silver sulfadiazine release test: this needs more details. Volumes, incubation times, source/composition of SEF, stirring applied, cummulative measurements, number of replicates, etc. Results and fig 13 suggest that incubation time is irrelevant for release from AgSD/Chitosan. After an immediate ‘burst’ of 8%, the release does not increase much further. Since release from the zoelite version does increase, longer incubation times should have been tested. Please use hours or days for the x-axis.
2.4.10 Antimicrobial activity: measuring the OD of bacterial cultures is not the proper technique to evaluate bactericidal effect. Use MIC (like in the indirect cytotoxicity assay) or time-kill assays vs extract concentration yielding CFU/ml. Even ZOI might be better. Why were different UV sterilization times used in 2.4.10 and 2.4.11? Number of replicates? Include error bars/standard deviation, significance and not a 3D graph in figure 14. The positive control is confusing and not useful since viability of the test samples is already expressed as % of control. A proper positive control would be the addition of for example AgNO3 in solution.
“Both films with AgSD were active against the proliferation of C. albicans and had a lower activity against the other bacterial strains.” This is not supported by the presented results. There is some inhibtion of growth of C. albicans but a reduction in ‘microbial density’ from 10^6 CFU/ml to approximately 4x10^5 CFU/ml after 12 h is rather insignificant. The last paragraph of section 3.3 is therefore irrelevant (no effect on bacterial density means no role of membrane composition).
2.4.11 Cytotoxicity: how long were cells incubated with Neutral red? Fixation and destaining with acetic acid and ethanol?
What is the explanation for the low viability at low extract concentrations of the zeolite formulation? Zeolite extract should be included, especially when “cytotoxicity data for zeolite are almost non-existent”. At present it is unclear whether zeolite is cytotoxic or (more likely) increases Ag release. Compare with for example “Hemocompatibility and cytocompatibility of pristine and plasma-treated silver-zeolite-chitosan composites”, Applied Surface Science, Volume 432, Part B, 2018, Pages 324-331. Of note: the undiluted extracts are cytotoxic but not bactericidal!
The statement “although our films have shown some cytotoxicity, it is likely that the in vivo test results are auspicious” (line 612) is guessing.
I fail to see the purpose of the last paragraph (line 618-624).
Conclusions:
“good antimicrobial activity”: this is not shown.
“AgSD-zeolite films are cytotoxic, although the conditions of the employed test are drastic”: explain drastic.
“under supervised conditions, and in a time/concentration dependent fashion the films would not exhibit any toxicity.” Please explain.
Reference 11 = reference 13
Author Response
Reviewer #2
This is an extensive study on the development and analysis of a AgSD containing film. The introduction needs to be re-written and proper references are required. At present it is not accurate and parts are not relevant. Especially the background regarding the impact of burn wounds on patients and the description of infection are improper.
The authors acknowledge the referee’s remark and introduction has been revised accordingly. Editing is marked in red. This section is now given as follows:
“A burn is a wound of any traumatic type that compromises the function of the epithelial tissue. It is considered a significant problem, not only for the gravity when acute, but also concerning its significant sequelae that may forever mark burned patients [1]. This type of injury is distinguished from others due to the high risk of colonization by pathogenic bacteria, the presence of large amounts of non-viable tissue, loss of a large quantity of water and blood, risk to remain open for extended periods of time until its complete healing, and frequently need to mobilize tissue for wound closure [2, 3].
Infection is one of the most frequent and severe complications of burn patients, being responsible for 75-80% of deaths worldwide. A local indication of infection includes blackened color of the burned area, evolution of partial necrosis to total necrosis, greenish coloration of the subcutaneous tissue, appearance of vesicles in healed lesions, quick detachment of the necrotic tissue, and the presence of phlogistic signs (hyperemia and edema) in areas close to burns [4].
Microbiota of healthy intact skin is characterized by the presence of some microorganisms, e.g., Staphylococcus epidermidis, Staphylococcus aureus, Streptococcus sp., Escherichia coli (perineum), Pseudomonas aeruginosa (underarms and inguinal regions) and Candida albicans [1, 5]. If not immune-compromised, these microorganisms do not represent an issue to intact skin, but they could represent a health problem in burned skin.
Severe burns are commonly treated with silver sulfadiazine 1% cream, which is applied onto the burned area, followed by protection with bandages composed of several layers of gauze and cotton. These dressings require frequent replacement because the bactericidal action of the cream lasts a maximum of 12 hours. The cream may even dry over time, with the consequent adherence of the dressing onto the wound surface, leading to pain, emotional trauma and considerable damage to the newly formed epithelium when the dressing is again removed [6, 7]. Another disadvantage regarding the use of the gauze dressings is the risk of almost occlusion of the wound and the accumulation of fluid below the dressing, which favors the proliferation of pathogens and delays the healing process [7, 8].
In this perspective, biopolymeric dressings, like those based on chitosan films, are promising to overcome these limitations. Moreover, chitosan exhibits antibacterial and antifungal activity [9], and also acts as an agent that assists in the natural blood coagulation, which serves as a protection for the nerves endings, reducing the pain [10].
The chemical structure of chitosan is similar to the structure of hyaluronic acid, which reinforces the indication of the use of this biopolymer as a healing agent. Besides, chitosan is capable of enhancing the role of inflammatory cells (e.g. polymorphonuclear leukocytes, macrophages) in promoting the cellular organization and in acting in the repair of large wounds [11]. Due to these properties, one of the most extensively medical applications for chitosan is the production of films to be used as burn wound dressing, hemostatic agent and material for surgical suture [12].
Zeolites are aluminosilicates widely used in the chemical industry. The high surface of zeolites can be exploited for the incorporation of molecules, like AgSD, for sustained release devices. The sustained release can even be enhanced by the inclusion of these zeolites into chitosan films [13]. The AgSD-impregnated zeolite can deliver AgSD directly to the wound and at a proper rate to act against microorganisms and promote fast healing. Tetrahedral structure of zeolite are arranged in rings, which are combined to form regular and uniform channels and cavities in which Ag is included and the release occurs over time [14].
The novelty of our work grounds on the development of an ideal dressing that combines the chitosan properties with the AgSD antimicrobial properties, together with the sustained release of sulfadiazine assisted by the zeolite. Chitosan film with silver sulfadiazine (zeolite free) was also prepared and evaluated for comparison. The resulting materials were physicochemically characterized, evaluated their antimicrobial properties and cytotoxicity profile in Balb/c 3T3 fibroblasts.”
Overall, grammar needs to be checked.
The manuscript has been revised for its use of English and grammar. It has been checked by an English native speaker before resubmission.
Gauze dressings are not occlusive but obstruct the wound from observation by clinicians.
The sentence has been edited as follows: “Another disadvantage regarding the use of the gauze dressings is the risk of almost occlusion of the wound and the accumulation of fluid below the dressing, which favors the proliferation of pathogens and delays the healing process [7, 8].”.
Page 3 please define “properly stored”.
We define “properly stored” as storing the film away from light and excessive humidity. The sentence has been edited as follows: “After drying, films were carefully removed from the dishes and properly stored in plastic containers to protect against direct light exposure and excessive moisture. Films with AgSD (zeolite free) were also prepared in the same way for comparison.”.
What is the added value of figure 1?
Figure 1 has been removed as suggested.
2.4.9 Silver sulfadiazine release test: this needs more details. Volumes, incubation times, source/composition of SEF, stirring applied, cummulative measurements, number of replicates, etc. Results and fig 13 suggest that incubation time is irrelevant for release from AgSD/Chitosan. After an immediate ‘burst’ of 8%, the release does not increase much further. Since release from the zoelite version does increase, longer incubation times should have been tested. Please use hours or days for the x-axis.
We acknowledge the referee’s remark and have included these details in the manuscript. This section is now given as follows: “To evaluate the release profile of AgSD from chitosan films, a simulated exudate fluid (SEF - 142 mmol/L sodium ions and 2.5 mmol/L calcium ions) representing the salt concentrations observed in wound fluids was used as medium and serum at a controlled temperature of 37°C. Release tests were performed using 4 x 4 cm film samples (average mass 0.15 g). Erlenmeyer flasks with the film samples and 100 mL of SEF solution were placed on an orbital shaker (Fisherbrand™, Fisher Scientific, Waltham, MA, USA) and kept under constant stirring throughout the assay to decrease the risk of mass convection resistance. Aliquots of 1.5 mL were taken at predetermined times over a total test time of 48 hours. The analysis was performed in duplicate and the silver released was analyzed by atomic absorption spectroscopy (AA SpectrAA 50B-Varian, USA) using a hollow cathode lamp (λ = 328 nm) and a mixture of air and acetylene generated the flame.”. We made the changes on the x axis. It would really be necessary to test the zeolite film for a longer time (beyond 48h). However, one can clearly see the trend of a sustained release of silver for a longer period of time; it the present work we set the release testing up to the 48 hours.
2.4.10 Antimicrobial activity: measuring the OD of bacterial cultures is not the proper technique to evaluate bactericidal effect. Use MIC (like in the indirect cytotoxicity assay) or time-kill assays vs extract concentration yielding CFU/ml. Even ZOI might be better.
We acknowledge the referee’s remark and we will certainly take these suggestions into account over the course of the planned experiments. For the present work, the suggesting tests are beyond the scope of this manuscript.
Why were different UV sterilization times used in 2.4.10 and 2.4.11?
Both were sterilized for 15 minutes. This information is given in section “2.4.10. Antimicrobial activity of chitosan films” as follows: “Antimicrobial activity analyses against common human pathogens, namely, Escherichia coli (ATCC 8739), Staphylococcus aureus (ATCC 6538), Pseudomonas aeruginosa (ATCC 9027) and Candida albicans (ATCC 10231), were carried out with the AgSD-loaded films. The strains were standardized using the technique of series dilution. Analyses were carried out by the inoculation of 106 CFU/mL of each strain in individual tubes containing 30 mL of TSB medium for the bacteria or 30 mL SDB medium for the yeast. The samples of films were cut into dimensions of 6 x 6 cm, and the sterilization were performed with UV light for 15 min each side. All sterilized samples were incubated in sterile test tubes. For the positive control, each tube was incubated with the microorganism only, whereas for the negative control the test tubes contained only TSB or SDB medium. The tubes were incubated at 37 °C for 12 h. Aliquots of 100 µL were transferred to flat-bottomed microplates. The microbiological growth was monitored by reading the optical density by an automatic microplate reader (Synergy HT Biotek, USA) at a wavelength of 620 nm.”, and in section “2.4.11 In vitro indirect cytotoxicity” as follows: “Balb/c 3T3 fibroblasts were grown in DMEM (Vitrocell Embriolife) supplemented with 10% fetal bovine serum (Vitrocell Embriolife). When fibroblasts reached 80% confluency, they were detached from the bottles by the action of trypsin (0.05% trypsin solution and 0.02% EDTA in phosphate buffer pH 7.2) and plated in 96-well plates (P96w) with 20,000 cells per well. Each sample film was cut and sterilized by exposure to a UV lamp for 15 min on both sides. After sterilization, the films were placed separately in sterile tubes containing serum (6 cm2/mL) for 72 h at 37oC to prepare the extracts. Extracts were filtered with membranes of 0.45 µm (nominal pore diameter), and seven dilutions were carried out. Each dilution was performed in sextuplicate. The medium was removed from the 96-well plate containing the fibroblasts, and the diluted extracts were added to fibroblasts cells. The plates were placed in a humid incubator (5% CO2 and 37oC) for 24 h. Then, extracts were removed, the wells washed twice with phosphate buffer pH 7.2 and a new medium with Neutral red was added for color development. The absorbance was measured in a microplate reader at a wavelength of 540 nm. The cell viability was calculated using the equation (04).”.
Number of replicates? Include error bars/standard deviation, significance and not a 3D graph in figure 14. The positive control is confusing and not useful since viability of the test samples is already expressed as % of control. A proper positive control would be the addition of for example AgNO3 in solution.
Figure 14 has been remade according to the suggestions and legend made comprehensive as follows: “Figure 14. Antimicrobial tests (cell viability of S. aureous, E. coli, P. aeruginosa and C. albicans) in liquid medium containing AgSD-Y/Chitosan film, AgSD/Chitosan film, Chitosan film or AgNO3. Results are the mean ± standard deviation (n = 3).”
“Both films with AgSD were active against the proliferation of C. albicans and had a lower activity against the other bacterial strains.” This is not supported by the presented results. There is some inhibtion of growth of C. albicans but a reduction in ‘microbial density’ from 10^6 CFU/ml to approximately 4x10^5 CFU/ml after 12 h is rather insignificant. The last paragraph of section 3.3 is therefore irrelevant (no effect on bacterial density means no role of membrane composition).
We removed the last paragraph.
2.4.11 Cytotoxicity: how long were cells incubated with Neutral red? Fixation and destaining with acetic acid and ethanol?
Thank you for your note. The ISO 10993, that we used as a guide, recommends that the biological evaluation of medical devices begins with the cytotoxicity assay in an established cell line (Balb 3T3/cA1), obtained from recognized repositories (ATCC). We have updated the information given in section 2.4.11. as follows: “Balb/c 3T3 fibroblasts were grown in DMEM (Vitrocell Embriolife) supplemented with 10% fetal bovine serum (Vitrocell Embriolife), as recommended by ISO 10993 [20].”. In this context we performed the viability test, from which we can also infer the proportion of dead cells in the study cell population, using Neutral Red as a viability dye, incubated for 3 hours. The procedure of desorbing of the cells uses a solution of 1%(v/v) of acetic acid, 49% (v/v) of ethanol and 50% (v/v) of water.
The guide recommends that we performed an extraction for 72 h at 37ºC. This fact could explain the cytotoxicity presented by the higher concentrations, in opposition of that related by the paper “Hemocompatibility and cytocompatibility of pristine and plasma-treated silver-zeolite-chitosan composites”, Applied Surface Science, Volume 432, Part B, 2018, Pages 324-331”. In this latter, the authors used an extraction of 24h at 37ºC, and diluted the samples prior to the contact with the cells. This fact could also explain why the extract is cytotoxic and not bactericidal, as the latter assay was performed for just 24 hours. From Figure 13, it is possible to note that the release of Ag is higher from AgSD-Y/Chitosan film than from AgSD/Chitosan film. This also could be noticed in the cytotoxicity assay, where the Ag release is probably the cause of the higher cytotoxicity presented by the higher concentration. In this context, the statement “although our films have shown some cytotoxicity, it is likely that the in vivo test results are auspicious” (line 612), is a natural conclusion.
We do plan nevertheless to continue the study by in-depth characterization of the zeolite cytotoxicity and genotoxicity to fulfill the lack of data in this field.
What is the explanation for the low viability at low extract concentrations of the zeolite formulation? Zeolite extract should be included, especially when “cytotoxicity data for zeolite are almost non-existent”. At present it is unclear whether zeolite is cytotoxic or (more likely) increases Ag release. Compare with for example “Hemocompatibility and cytocompatibility of pristine and plasma-treated silver-zeolite-chitosan composites”, Applied Surface Science, Volume 432, Part B, 2018, Pages 324-331. Of note: the undiluted extracts are cytotoxic but not bactericidal!
We acknowledge the referee’s remark; please refer to our answer above.
The statement “although our films have shown some cytotoxicity, it is likely that the in vivo test results are auspicious” (line 612) is guessing.
We acknowledge the referee’s remark; please refer to our answer above.
I fail to see the purpose of the last paragraph (line 618-624).
The paragraph was removed.
Conclusions:
“good antimicrobial activity”: this is not shown.
The word “good” was changed. Conclusions are now given as follows: “The polymeric chitosan films with dispersed AgSD or AgSD-impregnated zeolite demonstrated to be potential materials for the development of a new dressing. The impregnation process modifies the morphology of the starting zeolite. The addition of AgSD into the films did not result in substantial changes in the infra-red spectra, the XRD diffraction patterns, and the micrographs when compared to the pure chitosan films. In contrast, the addition of AgSD-Y zeolite resulted in less transparent films, with agglomerations of zeolites on the film structure. In these films, the presence of small agglomerates observed in the micrograph of the cross-section could be attributed to the dispersion of the zeolite powder into the chitosan solution and also to a possible physical interaction between the zeolite particles and the polymer chains detected in XRD diffractograms and infrared spectra. Moreover, the use of zeolites promoted a sustained release of the silver ions. The synthesized films showed some antimicrobial activity against the proliferation of C. albicans and a lower activity against bacterial strains. Gram-negative bacteria (P. aeruginosa and E.coli) were more susceptible to the action of silver than gram-positive bacteria (S. aureus) due to compositional differences in cell walls. The safety evaluation studies showed that although the AgSD-zeolite films exhibited a cytotoxicity profile at the tested conditions of 72 h of extraction, it is possible to adjust the user conditions and in a time/concentration dependent fashion so that the films would not pose any toxicological risk.”
“AgSD-zeolite films are cytotoxic, although the conditions of the employed test are drastic”: explain drastic.
“under supervised conditions, and in a time/concentration dependent fashion the films would not exhibit any toxicity.” Please explain.
We applied an extraction of 72 h at 37ºC, so if the dressing is changed at 24 hours or even at 48h we could expect a reduction of the cytotoxicity. To clarify this aspect, we have rephrased the last sentence of the conclusions as follows: “The safety evaluation studies showed that although the AgSD-zeolite films exhibited a cytotoxicity profile at the tested conditions of 72 h of extraction, it is possible to adjust the user conditions and in a time/concentration dependent fashion so that the films would not pose any toxicological risk.”.
Reference 11 = reference 13
We acknowledge the remark. References have been revised.
